# Temporal profiling of redox-dependent heterogeneity in single cells

Meytal Radzinski[1†], Rosi Fassler[1†], Ohad Yogev[1], William Breuer[2], Nadav Shai[3], Jenia Gutin[1,4], Sidra Ilyas[1], Yifat Geffen[1], Sabina Tsytkin-Kirschenzweig[5], Yaakov Nahmias[5], Tommer Ravid[1], Nir Friedman[1,4], Maya Schuldiner[3], Dana Reichmann[1]*

[1]Department of Biological Chemistry, The Alexander Silberman Institute of Life Sciences, Safra Campus Givat Ram, The Hebrew University of Jerusalem, Jerusalem, Israel; [2]Proteomics and Mass Spectrometry Unit, The Alexander Silberman Institute of Life Sciences, Safra Campus Givat Ram, The Hebrew University of Jerusalem, Jerusalem, Israel; [3]Department of Molecular Genetics, Weizmann Institute of Science, Rehovot, Israel; [4]School of Computer Science and Engineering, The Hebrew University of Jerusalem, Jerusalem, Israel; [5]Grass Center for Bioengineering, Benin School of Computer Science and Engineering, The Hebrew University of Jerusalem, Jerusalem, Israel

*For correspondence:
danare@mail.huji.ac.il

†These authors contributed equally to this work

Competing interests: The authors declare that no competing interests exist.

**Abstract** Cellular redox status affects diverse cellular functions, including proliferation, protein homeostasis, and aging. Thus, individual differences in redox status can give rise to distinct sub-populations even among cells with identical genetic backgrounds. Here, we have created a novel methodology to track redox status at single cell resolution using the redox-sensitive probe Grx1-roGFP2. Our method allows identification and sorting of sub-populations with different oxidation levels in either the cytosol, mitochondria or peroxisomes. Using this approach, we defined a redox-dependent heterogeneity of yeast cells and characterized growth, as well as proteomic and transcriptomic profiles of distinctive redox subpopulations. We report that, starting in late logarithmic growth, cells of the same age have a bi-modal distribution of oxidation status. A comparative proteomic analysis between these populations identified three key proteins, Hsp30, Dhh1, and Pnc1, which affect basal oxidation levels and may serve as first line of defense proteins in redox homeostasis.
DOI: https://doi.org/10.7554/eLife.37623.001

## Introduction

Cellular redox status has long been known to play a role in aging and cell functions, with increasing evidence linking aging-related diseases to changes in oxidation levels in a variety of organisms (*Bhat et al., 2017*; *Niedzielska et al., 2016*; *Paiva and Bozza, 2014*). Oxidative stress has been found to correlate with the generation of reactive oxygen species (ROS), protein aggregation, and apoptosis (*Madeo et al., 1999*; *Squier, 2001*), as well as a number of other physiological changes that ultimately occur in a heterogenic manner.

A series of useful probes developed in recent years have enabled assessment of redox changes within the cell (*Banach-Latapy et al., 2014*; *Lukyanov and Belousov, 2014*; *Meyer and Dick, 2010*). These are applicable in multiple organisms and can be targeted to sub-cellular compartments to report changes within specific organelles. While some of the probes are non-optical or lumines-cence-based, the majority are fluorescence-based, meant to identify changes in either signaling and regulatory oxidants, or redox homeostasis related redox couples such as glutathione (GSH/GSSG).

One prominent redox sensor is the redox-sensitive GFP variant (roGFP) fused to human Glutaredoxin 1 (Grx1) known as Grx1-roGFP2 which senses the glutathione redox potential, with additional variants targeting it to organelles such as the mitochondria or peroxisome. Other roGFP-based sensors specifically monitor changes in hydrogen peroxide ($H_2O_2$) levels, through use of thioredoxin peroxidases fused to roGFP molecules. These and other similar probes have been used to conduct real-time measurements of redox potentials in bacteria (*van der Heijden and Finlay, 2015*), *C. elegans* (*Braeckman et al., 2016*), plant (*Meyer et al., 2007*), and mammalian cells (*Dooley et al., 2004*), by monitoring differences in oxidative status under a range of diverse conditions.

Detection of roGFP redox-dependent fluorescence has generally been based either on imaging individual cells by microscopy, or by measuring the total fluorescence signals of cells in suspension by using plate readers. However, neither approach enables high spatiotemporal resolution in wide-scale tracking of cell to cell diversity, nor subsequent isolation of cells based on their redox status.

Over the last decade, numerous studies have pointed to the fact that populations of genetically identical cells are heterogeneous in their protein and gene expression (*Elowitz et al., 2002*; *Maamar et al., 2007*), exhibiting an array of differences in cellular behavior and in varying abilities to respond to changing environments (*Ackermann, 2015*; *Altschuler et al., 2010*; *Avery, 2006*). This cell-to-cell variability is considered to be one of the crucial features in the evolution of new survival strategies in fluctuating environments (*Altschuler et al., 2010*), antibiotic treatment (*Gefen and Balaban, 2009*), pathogen progression (*Avraham et al., 2015*; *Lieberman et al., 2014*) and other processes. However, the cell-to-cell heterogeneity of redox status within a population of synchronized cells (i.e. cells that have a shared chronological age) with an identical genetic background has not yet been explored.

Here, we developed a highly sensitive methodology based on the Grx1-fused roGFP2 redox sensor that uses flow cytometry to measure the redox state of individual cells within a heterogeneous *Saccharomyces cerevisiae* (henceforth referred to as yeast) population during chronological aging. Sorting of the yeast cells based on their oxidation status allows us to define the phenotypic, proteomic and transcriptomic profiles associated with the redox state of genetically identical cells of similar chronological age. We show that the proteomic and transcriptomic profiles of reduced and oxidized cells differ within a yeast population, in addition to corresponding changes in growth and cellular division. Comparative proteomic analysis identified three key proteins: the chaperone Hsp30, the helicase Dhh1, and the nicotinamidase Pnc1, which affect basal oxidation levels and might serve as first line of defense proteins in glutathione-dependent redox homeostasis.

We also demonstrate that although the ratio between the oxidized and reduced yeast subpopulations changes during chronological aging, the major features, including the transcriptome and proteome, remain linked to the redox status through 72 hr. By using cell imaging, we further show that there is a threshold of oxidation, above which the cell cannot maintain redox homeostasis (according to the glutathione-based probe). Finally, microscopic observations of budding cells show that once a mother cell is close to or above this threshold, it passes the oxidized state onto the daughter cell, which starts its life from a high, inherited oxidation level.

## Results

### Flow cytometry based methodology provides a highly accurate way to monitor the subcellular redox status of individual yeast cells

Cellular redox status has been suggested to be correlated with cell function and longevity (*Reverter-Branchat et al., 2004*). Measurements of cellular oxidation levels tend to be based on a global assessment of protein oxidation and the cellular redox status of the entire cellular population (*Kojer et al., 2012*). Intrigued by the possible heterogeneity of wild type yeast cells' oxidation levels, we developed a highly sensitive flow cytometry-based methodology to monitor and quantify the cellular redox status and sort out subpopulations of cells based on their oxidation levels.

To do so, we utilized the redox sensitive GFP variant Grx1-roGFP2 (*Figure 1A*) (*Morgan et al., 2011*), which has a characteristic fluorescence according to a redox-dependent conformational change (*Hanson et al., 2004*) that leads to an alternating peak intensity at 405 and 488 nm. In order to verify Grx1-roGFP2 expression patterns in our yeast cells, we measured the fluorescence spectrum of cells between 360 and 495 nm, following emission at 535 nm using plate reader analysis,

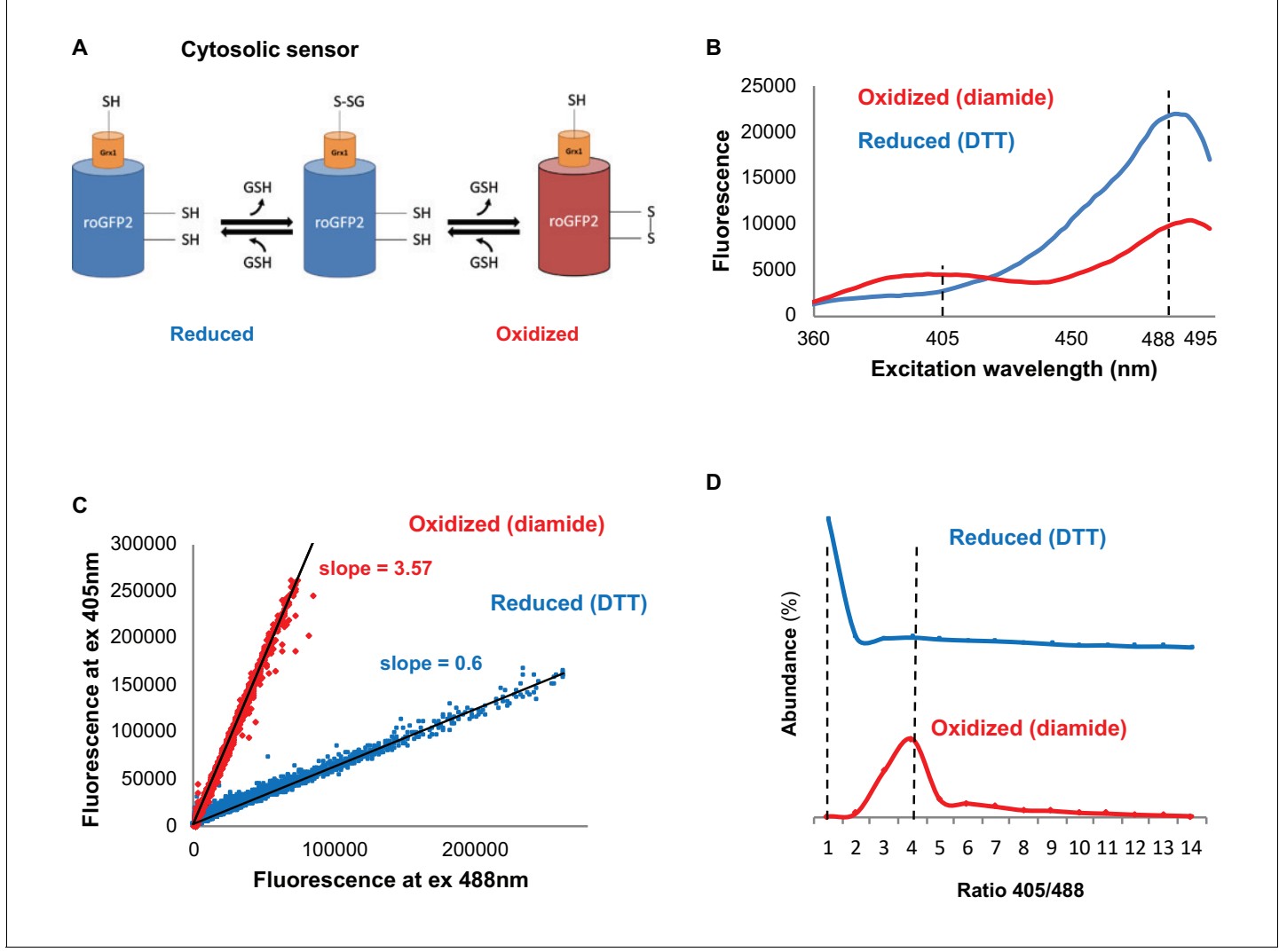

**Figure 1.** Monitoring subcellular redox levels in yeast cells using the Grx1-roGFP2 variants and flow cytometry. (A) Schematic of the Grx1-roGFP2 variant used in this study to monitor oxidation in the cytosol. Cellular GSSG reacts with the catalytic residues of the fused Grx1 which leads to oxidation of the modified GFP protein. (B) Fluorescence excitation spectra of Grx1-roGFP2 in fully reduced (blue) and fully oxidized (red) yeast cells. Emission followed at 510 nm. (C–D) Quantification of redox status of fully reduced (blue) and fully oxidized (red) cells using FACS. (C) Fluorescence of Grx1-roGFP2 at 510 nm obtained using excitation by 405mn and 488 nm lasers. (D) Distribution of the 405/488 nm ratios among fully reduced and fully oxidized cells.

DOI: https://doi.org/10.7554/eLife.37623.002

The following figure supplements are available for figure 1:

**Figure supplement 1.** Representation of FACS subpopulation gates.

DOI: https://doi.org/10.7554/eLife.37623.003

**Figure supplement 2.** Linear characterization of (A) peroxisomal and (B) mitochondrial 'oxidation gate'.

DOI: https://doi.org/10.7554/eLife.37623.004

under either oxidizing or reducing conditions (*Figure 1B*). In agreement with previous studies, we identified the alternating peak pattern at 405 and 488 nm (*Hanson et al., 2004*), which indicates the differences in ratio between 405 and 488 nm under oxidizing or reducing conditions.

To dissect cell-to-cell variation, we applied flow cytometry to quantify the ratiometric fluorescence intensity of individual cells in a population at 405 and 488 nm. First, we determined the degree of probe oxidation expressed in cells by measuring the differential Grx1-roGFP2 fluorescence under oxidative and reducing conditions (15 min treatment with 8 mM Diamide and 40 mM DTT, respectively) (*Figure 1C,D*), noting that the alternating peak pattern at 405 and 488 nm reflects

a modestly different ratio than that identified using a plate reader (*Figure 1B*) rooted in the differences between the instruments (i.e. sensitivity), mode of measurement and live-cell diversity. To focus on living cells within a wild type (BY4741), untreated population, we gated out the dead cells and identified Grx1-roGFP2 negative cells, leaving only cells that had a fluorescence intensity at both 405 and 488 nm. The advantage of using the ratiometric approach is that the outcome does not depend on probe expression, which can vary with age and/or treatments. The 405/488 nm ratios of reduced and oxidized cells were significantly different and could be characterized using linear fits with a slope of 3.57 for oxidized and 0.6 for reduced cells (the correlation coefficients ($R^2$) were 0.99 and 0.97 for oxidized and reduced cells, respectively) (*Figure 1C,D*). This approach enabled us to quantitatively determine a normalized value (OxD) which ranges from 0 (reduced) to 1 (oxidized), as described in the Materials and methods section, and to apply it to the quantification of the relative redox status of the cells.

The discrete difference between the oxidized and reduced states enabled us to create gates for oxidized and reduced cells (*Figure 1—figure supplement 1*), such that we could sort and characterize yeast subpopulations according to their redox status. This approach was also applied to organelle-specific redox sensors, such as the mitochondrial Grx1-roGFP2-Su9 (*Kojer et al., 2012*) and the peroxisomal Grx1-roGFP2-SKL (*Elbaz-Alon et al., 2014*), revealing a similar redox sensitivity (*Figure 1—figure supplement 2*).

## Defining redox-dependent heterogeneity in a chronologically aging population

The association between chronological aging and cellular oxidation has been explored in previous studies, with research indicating that older cells in a culture undergo changes in their cellular redox status (*Herker et al., 2004*; *Brandes et al., 2013*). To examine the ability of our flowcytometry-based method to monitor the age-related oxidation in yeast cells, we first used the cytosolic Grx1-roGFP2 redox sensor to evaluate the average glutathione-dependent cellular redox changes of the wild type yeast strain over four days of growth in standard medium enriched with casein digest under aerobic conditions (as described in Materials and methods). As expected, cellular oxidation increased with chronological age (*Figure 2A*). Interestingly, when using the mitochondrial matrix sensor (Grx1-roGFP2-Su9) on the same wild type strain, we identified a smaller age-dependent difference in mitochondrial versus cytosolic oxidation, suggesting a robustness of mitochondrial antioxidant behavior during chronological aging (*Figure 2B*).

Although the average cytosolic OxD clearly shows an age-dependent increase in oxidation, it does not give an indication of the potential heterogeneity among cells of the same age. When we examined the distribution of the fluorescence intensity ratios of Grx1-roGFP2 (at 405 and 488 nm) over four days of growth under chronological aging conditions (24, 48, 72, 96 hr), we found a clear bi-modal distribution of reduced and oxidized subpopulations (*Figure 2C*). As the culture aged, the fraction of reduced cells decreased while the fraction of oxidized cells increased. This fraction comprised of higher 405/488 nm ratios as compared to the distribution shown in *Figure 1D*, reflecting the natural, average signal in an untreated cell which largely corresponds with the 'oxidized' gate, yet has a unique expression pattern. Notably, the largest shift from reduction to oxidation occurs after 48 hr of growth. Thus, we were able to monitor the time-dependent shift from reduced to oxidized subpopulations as the yeast cells aged. However, we noted that a majority of cells remained reduced over time, suggesting that despite the culture being in the stationary phase, some cells nonetheless undergo replication and produce reduced cells.

It is important to note that while the average OxD serves as a useful metric by which to analyze yeast populations by flow cytometry, its value diminishes somewhat on an individual, cell-to-cell scale. Unlike whole population averages or microscopy-based individual-cell assessments, the OxD value is not as sensitive to smaller deviations for individual cells within a population (*Figure 2—figure supplement 1*). This is largely due to the difficulty in normalization of individual cells using fluorescence values derived from an average oxidized/reduced population, rather than comparison to a single cell (which would not be mathematically representative). As a result, this normalization masks cell-to-cell variability, resulting in inflated scales (e.g., 150% oxidation). Despite the limitations of using OxD normalization for individual cells of this kind, the distribution shape remains constant.

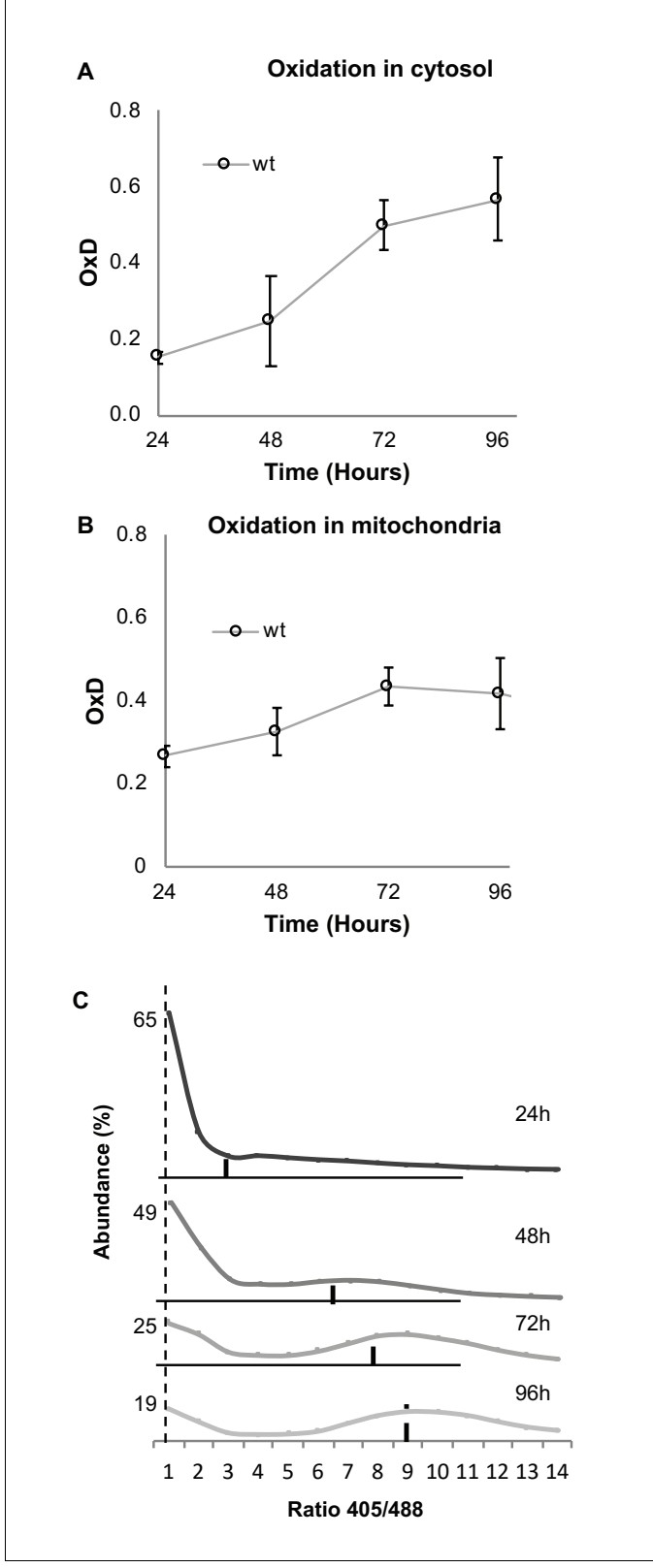

**Figure 2.** Both cytosolic and mitochondrial oxidation levels increase with age and indicate the emergence of oxidized and reduced cellular subpopulations. Cells expressing cytosolic Grx1-roGFP2 (**A**) or mitochondrial Grx1-roGFP2-Su9 (**B**) probes in wild type cells were grown for four days and their oxidation level monitored. In each experiment 10,000 cells were measured. Shown are oxidation changes over four days in both the cytosol and

*Figure 2 continued on next page*

*Figure 2 continued*

mitochondria. (C) Bi-modal distribution of ratios of fluorescence intensities obtained at 405 and 488 nm in yeast samples of different ages (24, 48, 72, and 96 hr), pursuant to *Figure 1D*. A peak at ratio ≤1 represents the reduced subpopulation, while ≥6 represents the oxidized subpopulation.

DOI: https://doi.org/10.7554/eLife.37623.005

The following figure supplement is available for figure 2:

**Figure supplement 1.** Representative examples of OxD distribution curves at 24, 48, and 72 hr, in contrast to 405/ 488 nm ratios shown in *Figure 2*.

DOI: https://doi.org/10.7554/eLife.37623.006

## Growth differences between the reduced and oxidized subpopulations

Our original expectation, based on the current consensus, was of a gradual increase in the cellular oxidation status of all cells, thus we were intrigued by the observed bi-modal distribution. We decided to characterize in greater detail the reduced and oxidized subpopulations of aging cells, utilizing the previously described predefined flow cytometry gates to isolate these subpopulations by fluorescence activated cell sorting (FACS). The sorted cell subpopulations (reduced and oxidized) were divided into three samples, in order to: monitor growth rate, quantify division events, and conduct proteomic and transcriptomic analyses. This process was repeated at least three times in order to provide biological replicates. Due to the low number of oxidized cells obtained after 24 hr growth, we focused on cultures aged 48, 72 and 96 hr.

The subpopulations of the same chronological age (i.e. cultures grown for 48 or 72 hr before sorting), displayed characteristic differences in their recovery growth rate, whereas the reduced cells exhibited a significantly faster minimal doubling time (MDT) at 48 and 72 hr relative to the oxidized population of the same age (*Figure 3A*). These differences are indicative of an apparent impairment in the oxidized population's ability to rapidly divide and grow as a culture, rather than any inherent inability to enter the logarithmic growth stage (as evidenced by the similar lag phase lengths, *Figure 3B*). After 96 hr, both subpopulations had ostensibly similar doubling times following longer lag phases than younger cells (*Figure 3B*), consistent with the fact that these were relatively old cells with decreased vitality, particularly among the oxidized cells.

The differences in the growth rates were further supported by a difference in the average number of division or bud scars per cell (*Figure 3C*). The oxidized population displayed on average 3–4.5 times more division scars than the reduced population, which typically displayed cells with no more than one scar (*Figure 3C*), indicating that the oxidized subpopulation had undergone multiple divisions in contrast to the reduced subpopulation.

Furthermore, the oxidized subpopulation displayed a far lower budding rate relative to the reduced subpopulation, in which approximately 30% of cells at 48 hr were observed to be in the process of budding (*Table 1*). These results suggest a correlation between the degree of cellular oxidation and division events in the stationary stage.

## Reduced subpopulation has an increased oxygen consumption rate

To evaluate differences in the metabolic capacity between the reduced and oxidized subpopulations, we measured the oxygen consumption rate (OCR) and extracellular acidification rate (ECAR), a surrogate measure of lactate production, for each isolated subpopulation. The number of cells was equal in each sample and three biological replicates were analyzed. The OCR of the oxidized cells was consistently 1.8–2 fold lower than the reduced population (*Figure 4A*), while their ECAR rate was 2–2.8 fold higher (*Figure 4B*), suggesting a strong shift from oxidative phosphorylation to glycolysis reminiscent of the Warburg effect in the oxidized subpopulation. Decreased oxygen consumption is strongly correlated with decreased cell growth of the oxidized subpopulation as compared to the reduced subpopulation.

## Cellular oxidation correlates with replicative aging but does not exclusively depend on it

Intrigued by the differences in replication between the two subpopulations, we sought to investigate the correlation between age and redox status more closely, with a focus on the replicative life span

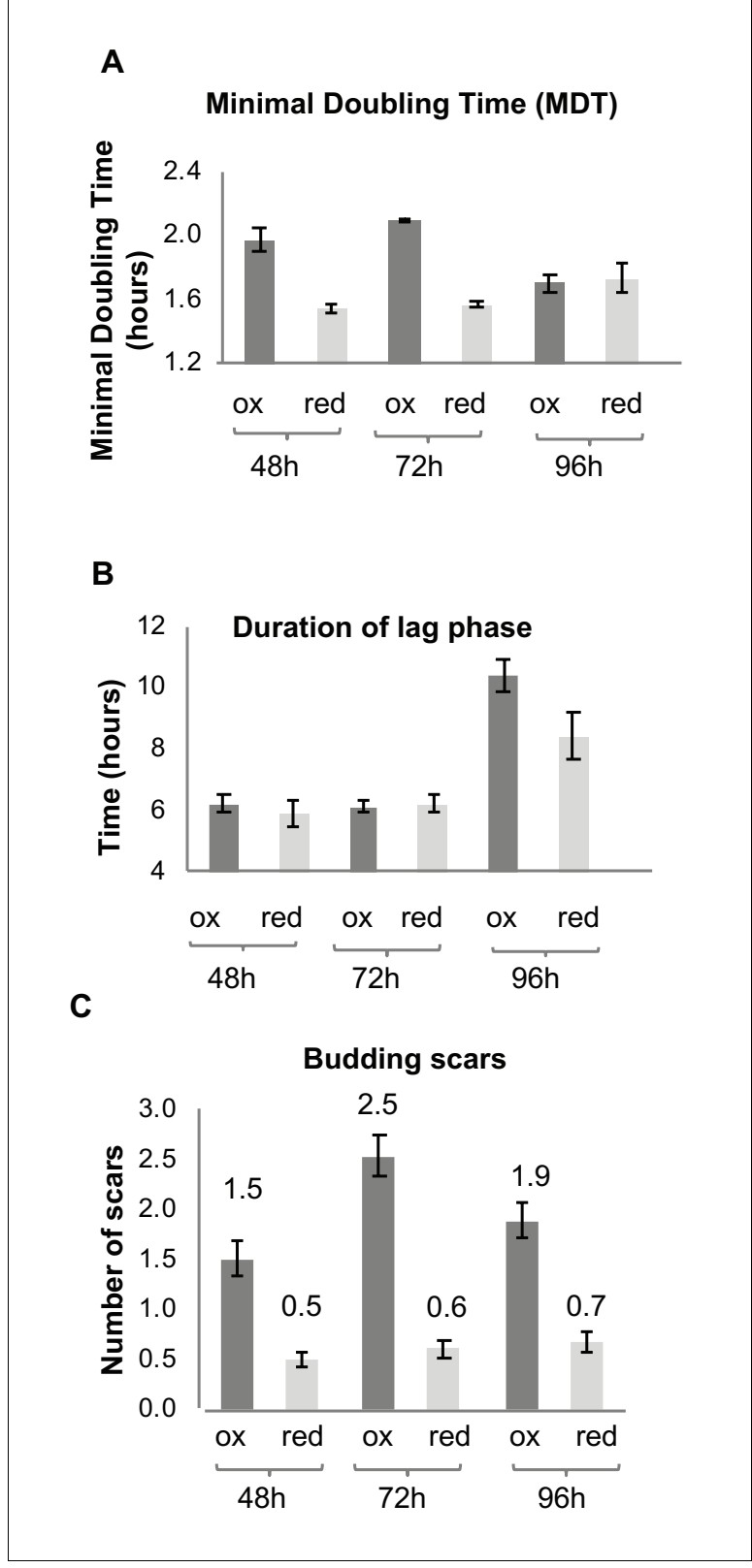

**Figure 3.** Growth and division of reduced and oxidized subpopulations of yeast cells after sorting by FACS. (**A**) Differences in minimal doubling time between the sorted oxidized and reduced subpopulations at different time points (48, 72, and 96 hr) measured by $OD_{600}$ in a plate reader. (**B**) Corresponding duration of the lag phase
*Figure 3 continued on next page*

*Figure 3 continued*

during recovery growth, measured by plate reader. (**C**) Corresponding differences in average number of budding scars counted in sorted oxidized and reduced subpopulations, as assessed by confocal microscopy.

DOI: https://doi.org/10.7554/eLife.37623.007

(RLS) aging model. Using this model, we tracked cell division events according to changes in budding scars and compared the redox status of aged cells.

Using the Mother Enrichment Program (*Lindstrom and Gottschling, 2009*), we grew a population of Grx1-roGFP2-positive cells with a selection against daughter cells and a favorable environment for continued division by mother cells and thus enrichment of highly divided mother cells (*Figure 5A*). To correlate between replicative aging and chronological aging the samples were then grown for up to 72 hr, and analyzed using flow cytometry at two stages: early (24 hr) and late growth (72 hr). Bud scars were labeled using Texas Red-X conjugate of wheat germ agglutinin (WGA) lectin, with flow cytometry analysis centered around the increased fluorescent signal following bud-scar enrichment (as described in Materials and methods). Samples were thus divided into two additional subpopulations, based on approximate bud-scar counts (high and low) as shown in *Figure 5B,E*. In parallel, oxidation of the cells was evaluated by the Grx1-roGFP2 sensor for each subpopulation.

During early growth (after 24 hr incubation) (*Figure 5B*), two subpopulations emerge with regards to bud scar count, with one demonstrating a stronger expression of the lectin dye associated with a higher bud scar count (labeled as p1 in *Figure 5B*) and one with lower bud scar counts (labeled as p3 in *Figure 5B*). Among the GFP-positive cells the subpopulation of cells with a lower bud scar count has a ratio that closely resembles early growth under standard conditions as described earlier, with a strong peak distributed around reduced ratios, and a very minor subpopulation in the oxidized range (6.5% in the representative figure) (*Figure 5C*). The high bud scar subpopulation, meanwhile, clearly falls into the oxidized range (87.8% in the representative figure), with distribution of the corresponding 405/488 nm ratios indicating a wide peak around ratios higher than 4 (*Figure 5D*). However, a small minority of the subpopulation is nonetheless reduced, despite having undergone multiple divisions. Note, that a small group of cells fall slightly beyond the definitions of the defined gates, or has no GFP signal, hence percentages presented in the figures do not sum to exactly 100%.

During late growth (after 72 hr incubation), a marked shift occurs towards cells with an increased number of bud scars (p1, *Figure 5E*), with a smaller subpopulation of cells remaining with fewer scars (p3, *Figure 5E*) (i.e., cells that have divided fewer times during growth). Similar to early growth, the subpopulation that has undergone multiple divisions is overwhelmingly oxidized (87.6%), with a very small group of reduced cells (3%) (*Figure 5G*). This latter group is of particular interest, as it represents cells that have undergone multiple divisions and replicative events, yet have maintained a reduced environment.

Interestingly, among the subpopulation with a lower bud scar count, we detect two distinct populations of cells with reduced and oxidized ratios. Unlike the early growth sample, a larger fraction of young mothers underwent oxidation (36.4 vs 6.5%), suggesting a transition from the largely reduced population at early growth to the highly oxidized population at late growth (*Figure 5F* vs *Figure 5C*). Thus, cellular oxidation is not embedded solely in number of divisions, but also in the duration of the stationary phase (chronological aging).

Together, these results strengthen the correlation described earlier between the cellular redox state and cell divisions, here identifying a link between cells that have undergone multiple divisions and a strong tendency towards oxidation, though not an exclusive one. We observe that despite

**Table 1.** Number of budding events

| Growth (h) | Oxidized (%) | Reduced (%) |
| --- | --- | --- |
| 48 | 10 | 30 |
| 72 | 9 | 21 |
| 96 | 6 | 15 |

DOI: https://doi.org/10.7554/eLife.37623.008

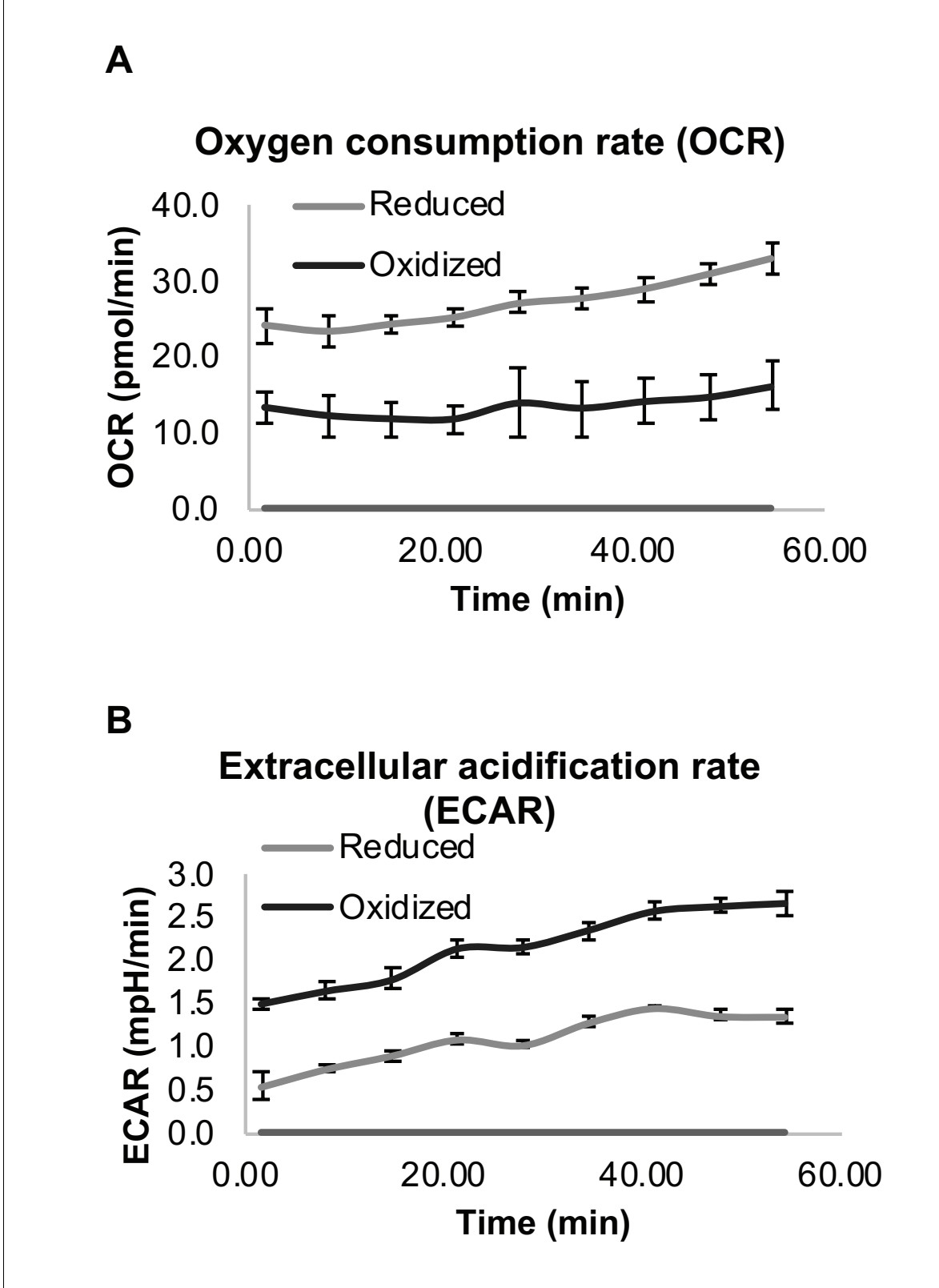

**Figure 4.** Differential respiration profile of reduced and oxidized subpopulations after sorting by FACS. (**A**) Normalized oxygen consumption rate (OCR) of reduced and oxidized yeast populations. (**B**) Normalized extracellular acidification rate (ECAR) of reduced and oxidized yeast populations. Three biological replicates with an identical number of cells were analyzed for each measurement.

DOI: https://doi.org/10.7554/eLife.37623.009

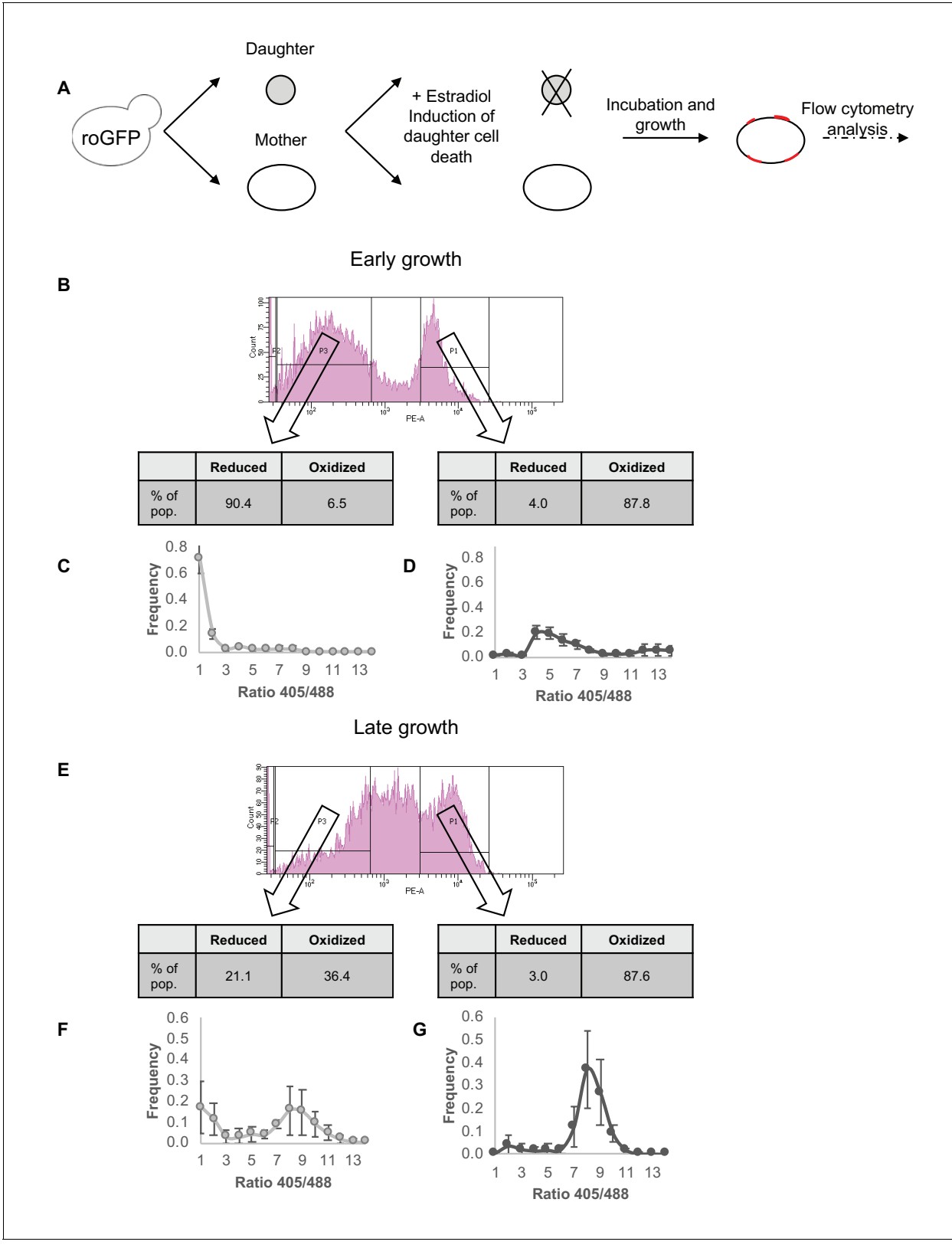

**Figure 5.** Correlation between redox status and replicative aging. (**A**) Schematic of mother enrichment program in yeast, using estradiol induction for daughter cell death, growth, and flow cytometry analysis of both bud scar counts and oxidation level. (**B**) Representative histogram of low and high bud scar subpopulations during early growth. Within the histogram, p1 represents the subset of the population with a higher number of bud scars, while p3 a lower. In table: The fraction of cells belonging to the Grx1-roGFP2-determined reduced or oxidized subpopulations changes according to bud scar

*Figure 5 continued on next page*

*Figure 5 continued*

count, with an enrichment for oxidized cells within the more highly divided subpopulation. (C) Distribution of the ratios of fluorescence intensities obtained at 405 and 488 nm among the low bud scar subpopulation at early growth, with a strong peak in reduced ratios. (D) Distribution of the ratios of fluorescence intensities obtained at 405 and 488 nm among the high bud scar subpopulation at early growth, with a wide peak around oxidized ratios of varying degrees. (E) Representative histogram of low and high bud scar subpopulations during late growth. Within the histogram, p1 represents the subset of the population with a higher number of bud scars, while p3 a lower. At late growth, the entire population displays an enrichment for oxidized cells, while the low bud scar subpopulation nonetheless contains a higher fraction of reduced cells. (F) Distribution of the ratios of fluorescence intensities obtained at 405 and 488 nm among the low bud scar subpopulation at late growth, with a bimodal distribution between oxidized and reduced ratios. (G) Distribution of the ratios of fluorescence intensities obtained at 405 and 488 nm among the high bud scar subpopulation at late growth, with a strong peak around oxidized ratios.

DOI: https://doi.org/10.7554/eLife.37623.010

undergoing many replications, a small subpopulation of the mother cells remains 'eternally reduced', while there is a complementary subpopulation of cells that have divided relatively few times that begin life in an oxidized state. This suggests a strong, though not absolute, correlation between replicative aging and redox state. However, it is worth noting that these exceptionally rare cells (in both cases) are representative of the replicative aging model, rather than a chronologically aged culture, in which there appears to be greater turnover in terms of daughter cells and constant replication (including during the stationary phase).

Furthermore, we observe a shift in the redox ratio between the subpopulation of highly divided cells at early growth as compared to late growth, while the degree of cell divisions itself (as identified by Texas Red-X intensity) does not change. This points to additional redox changes unrelated to division and replicative events, correlated instead with chronological aging.

## Comparison of proteome profiles of the reduced and oxidized subpopulations

In order to further characterize the oxidized and reduced subpopulations of cells, we utilized mass spectrometry to profile the proteome of these subpopulations after 48 and 72 hr of growth. Three biological replicates of each sample group were collected, lysed, trypsin-digested and analyzed by liquid-chromatography-mass spectrometry. Using MaxQuant analysis and a stringent filter, we identified 3389 proteins, of which 1019 were identified in all four subpopulations at least twice (*Supplementary file 2*). According to the Uniprot-based cellular localization annotation, the majority of our identified proteins were cytosolic (51%), where the remaining proteins were associated with the mitochondria (16%), nucleus (19%), ER (6%), Golgi vesicles (4%), vacuole (3%), stress granules, P-bodies (0.7%) and peroxisomes (1%) (*Supplementary file 2*, *Figure 6—figure supplement 1*). This is consistent with general distributions of yeast proteins (47% cytosol, 15% mitochondria and 13% ER and secretory vesicles [*Kumar et al., 2002*]) suggesting that no dramatic expansion or shrinking of an organelle occurred during the experiment.

To obtain further insights into the global changes in the proteome profile of the oxidized and reduced subpopulations, we utilized label-free quantification (LFQ) (*Cox and Mann, 2008*) to compare expression of the identified proteins between the sample groups. Clustering of average LFQ intensities for all sample groups revealed that the protein expression profiles were similar between groups with a similar redox status, regardless of the sample age (48 or 72 hr) (*Figure 6A* and *Figure 6—figure supplement 2*). The two largest clusters, comprising more than 600 proteins, showed that the differential expression between oxidized and reduced subpopulations remained consistent for over two days (*Figure 6A*, clusters 3 and 10, *Supplementary file 2*). Annotation enrichment analysis of the proteins comprising these two clusters suggested that the reduced cells (48 and 72 hr) had an increased presence of proteins involved in energy production, including mitochondrial proteins, protein biogenesis and protein degradation (*Figure 6B*). This corresponds with our previous finding that the reduced subpopulation had an elevated growth rate and was in the process of division. The oxidized cells had a distinct subset of proteins regulating protein folding and redox homeostasis, alongside oxidoreductases and NAD binding proteins (*Figure 6B*).

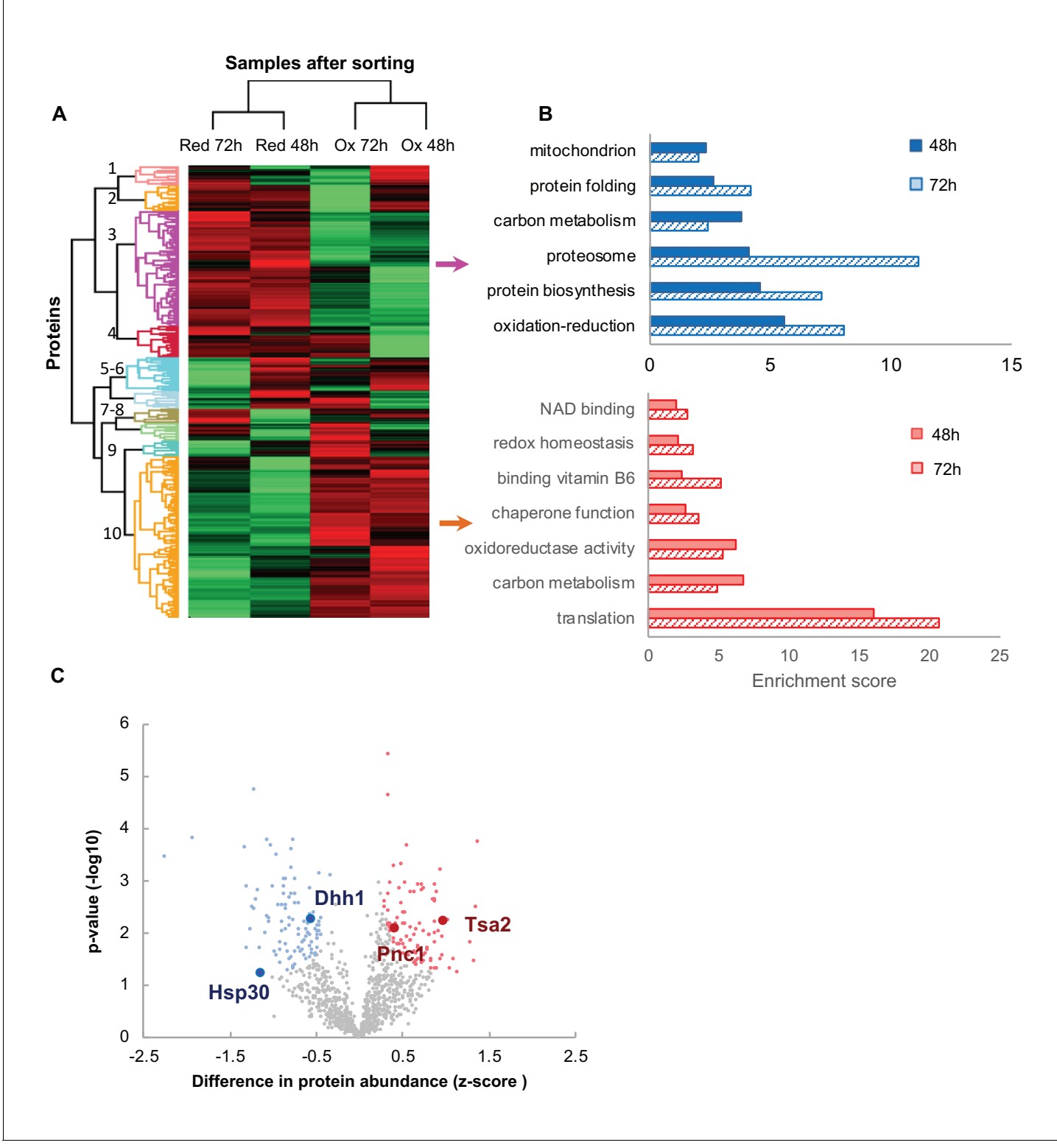

**Figure 6.** Proteomic analysis and functional enrichment analysis of the reduced and oxidized subpopulations. (**A**) Hierarchical clustering of proteins identified in cells sorted from cultures at different time points (48 and 72 hr); downregulated proteins are in green, up-regulated are in red. (**B**) Functional enrichment analysis of the two largest differentially expressed clusters (3 and 10, representing enriched functions in the reduced and oxidized subpopulations, respectively) at 48 and 72 hr (solid fill and slashes, respectively). (**C**) Volcano plot of differentially expressed proteins between the reduced and oxidized subpopulations. Significantly expressed proteins are labeled in blue (increased expression in reduced) and red (increased expression in oxidized), according to an FDR of 0.05 and a fold change greater than 2.

*Figure 6 continued on next page*

*Figure 6 continued*

DOI: https://doi.org/10.7554/eLife.37623.011

The following figure supplements are available for figure 6:

**Figure supplement 1.** Cellular localization annotation of all identified proteins (in at least two repeats within all four subpopulations).

DOI: https://doi.org/10.7554/eLife.37623.012

**Figure supplement 2.** Hierarchical clustering of the protein abundances.

DOI: https://doi.org/10.7554/eLife.37623.013

**Figure supplement 3.** Functional enrichment of differentially expressed proteins (FDR < 0.05) in the oxidized (red) and reduced (blue) subpopulations (corresponding to the volcano plot in *Figure 6*).

DOI: https://doi.org/10.7554/eLife.37623.014

## Differences in the levels of specific proteins between the reduced and oxidized cell subpopulations

One of the big unanswered questions arising from our data is how cells of the same chronological age co-exist at different oxidation levels. To identify potential key proteins mediating the redox status of cells, we applied a stringent T-test analysis (false detection rate [FDR]=0.05) to compare protein expression in the oxidized and reduced populations of the same age. We identified a subset of 199 proteins showing statistically significant different expression profiles between the reduced and oxidized populations after 48 hr growth and 149 proteins after 72 hr growth (*Figure 6C*, *Supplementary files 3* and *4*). Annotation enrichment analysis showed a similar functional classification for clusters 3 and 10 in *Figure 6* (*Figure 6—figure supplement 3*), suggesting that the reduced cells were more metabolically active, expressing higher levels of mitochondrial and carbon metabolism proteins than the oxidized subpopulation. We hypothesized that clustered proteins might belong to a common pathway which would be significantly up- or down-regulated in a redox-dependent manner. Hence, to identify potential protein-protein interactions within the subset of differentially expressed proteins, we used the STRING protein interaction database (*Szklarczyk et al., 2017*) to characterize the possible interaction networks between these proteins. We obtained numerous clusters of potential interactions between our identified, differentially expressed proteins, associated with protein translation, mitochondrial activity, and stress response (*Figure 7—figure supplement 1*).

Specifically, the proteome of the oxidized cells contained significant upregulation of proteins induced by a variety of stresses (Gre1, Sro9, Hsp150), as well as proteins directly involved in redox homeostasis (thioredoxin peroxidase Tsa2, Tma19). Interestingly, the oxidized cells displayed an upregulated expression of ribosomal proteins, in which we identified 21 proteins belonging to 40 and 60S ribosomal subunits, suggesting relatively elevated levels of ribosomes under oxidizing conditions. Conversely, the reduced cells had high levels of proteins related to protein biosynthesis, tRNA-associated proteins, folding factors and chaperones. Hence we speculate that the oxidized cells may accumulate 'non-functional' ribosomes whereas the reduced cells were in the process of protein biosynthesis and energy production.

Moreover, elevated levels of cytosolic catalase T (Ctt1) and mitochondrial glutaredoxin (Grx2) were detected in the reduced subpopulation, likely preserving the reduced state of these cells. We also identified the upregulation of several proteins that have been linked to increased longevity, such as Dld1 and Gut2 (*Easlon et al., 2008*), implying that the redox-dependent expression of these proteins may play an active role in cellular lifespans. Intriguingly, the reduced cells showed increased expression levels of proteins associated with stress granules and stress response, such as stress-induced heat shock proteins: Hsp30, Hsp70-type chaperone Ssb1, its ATP exchange factor, Sse2, disaggregase Hsp78, as well as the RNA metabolism proteins: DEAD-box helicase Dhh1, pre-mRNA cleavage protein Hrp1, translational factor Prt1, exonuclease Xrn1 and others (*Figure 7—figure supplement 1*, *Supplementary files 3* and *4*). One intriguing possibility is that these proteins serve as a first line of defense against oxidation, such that their expression prevents further cellular oxidation.

## The potential impact of Hsp30, Dhh1 and Pnc1 on cellular redox status

To verify our proteomic analysis and identify potential first line of defense redox proteins, we focused on four selected candidates whose expression significantly increased either in the reduced

cells – heat shock protein 30 (Hsp30) and helicase Dhh1 - or in the oxidized cells - thioredoxin peroxidase Tsa2 and nicotinamidase Pnc1. All four proteins have been found to be associated (based on the STRING database) with other significantly differentially expressed proteins identified in our proteomic analysis (*Figure 7—figure supplement 1A–C*).

To examine the contribution of these selected proteins to the average glutathione-dependent cellular redox status during 72 hr growth, we measured redox levels using the cytosolic Grx1-roGFP2 sensor in the wild type strain, as well as in four knockout strains: Δ*tsa2*, Δ*dhh1*, Δ*pnc1* and Δ*hsp30*. Notably, we found that Δ*dhh1* and Δ*hsp30* were significantly more oxidized than the wild type over 72 hr (*Figure 7A*, p-values of the t-test are in *Supplementary file 6*). Both proteins were found to be upregulated in the reduced subpopulation, suggesting that they may play a role in redox regulation pathway mediation. Consistent with previously published studies, the Tsa2 knockout had no significant effect on cellular oxidation (*Park et al., 2000*) (*Figure 7A*, *Supplementary file 6*). However, deletion of the nicotinamidase Pnc1, which was found to be upregulated in the oxidized cells, led to a more reduced environment than in wild type cells, specifically after 48 hr of growth (*Figure 7A*, *Supplementary file 6*). This corresponds with our proteomic analysis and also suggests that Pnc1 may be a potential effector of redox homeostasis.

In addition to their contribution to the average cellular redox status, we examined the impact Hsp30, Dhh1, and Pnc1 have on individual cell growth. Δ*dhh1*, Δ*pnc1* and Δ*hsp30* deletion strains were grown to 48 hr and monitored using confocal microscopy (as described in Materials and methods). Here, we found Δ*dhh1* to be under considerable stress from the onset, with a weaker fluorescent expression and a significantly higher rate of apoptosis as compared to the other strains (*Mazzoni et al., 2003*). Growth rate and replication were similarly stunted in comparison with the other strains (*Duan et al., 2013*; *Marek and Korona, 2013*), in addition to differences in the general appearance of the cells (which were larger and more oval) (*Tseng-Rogenski et al., 2003*) (*Figure 7B*). Furthermore, we observed a distinct increase in vacuole size in the Δ*dhh1* strain, suggesting Dhh1's role in regulating vacuolar traffic (*Chan and Marshall, 2014*). In comparison, neither Δ*pnc1* nor Δ*hsp30* display any change in replication or growth (*Figure 6—figure supplement 3*), indeed surpassing the wild type in number of cells after 6 hr growth (*Videos 1–4*). Despite no seemingly beneficial impact on their growth rate, both Δ*pnc1* and Δ*hsp30* display some phenotypic change in regards to size and texture. Δ*pnc1* cells on average present as slightly smaller than the wild type, while Δ*hsp30* are on average slightly larger than the wild type, yet reproduce at a similar rate. Both Δ*pnc1* and Δ*hsp30*, meanwhile, display small, granule-like clusters within the cell, which may be identified as yeast lipid droplets (*Wang et al., 2014*). Together, these results suggest that deletion of the redox-related proteins affects more than just the average redox status, and is worth further investigation.

We then examined the distribution of cellular oxidation levels within the knockout strain populations, to evaluate the potential effect of Dhh1, Hsp30 and Pnc1 on redox-dependent heterogeneity during chronological aging (*Figure 7C–F*). For comparison, we used the Δ*tsa2* variant that showed wild type-like OxD levels (*Figure 7A*). As expected, we observed a gradual shift from the lower (reduced) ratios to more oxidized ratios in the wild type-like Δ*tsa2* (*Figure 7C*). Interestingly, Δ*dhh1* displayed a strong shift towards higher, more oxidized 405/488 nm ratios from early growth (24 hr), with a clear collapse at 72 hr alongside a significant decrease in living cells (*Figure 7D*). Conversely, Δ*pnc1* remained largely the same over time, with little change in the dominant fraction of the reduced subpopulation (*Figure 7E*). Finally, Δ*hsp30* displayed a unique bimodal distribution that changed little over time, explaining its relatively unchanged, moderately high OxD values (*Figure 7F*).

To investigate the origin of increased oxidation in the Δ*hsp30* strains, we monitored oxidation changes in mitochondria (*Figure 7—figure supplement 3A,C*) and peroxisome (*Figure 7—figure supplement 4A,D*) using the Grx1-roGFP2-Su9 and Grx1-roGFP2-SKL sensors, respectively. Interestingly, Hsp30 deletion resulted in the unique bi-modular distribution of the 405/488 nm ratios in the mitochondria but not in the peroxisome (*Figure 7—figure supplement 3C* vs *Figure 7—figure supplement 4D*). It is tempting to speculate that Hsp30 affects the redox synchronization in the mitochondria and cytosol. This potential role poses a fascinating question for future exploration.

Moreover, mitochondrial and peroxisomal oxidation was characterized for Δ*pnc1* and Δ*tsa2* mutants, revealing no significant changes in distribution of the 405/488 nm ratios (*Figure 7—figure supplement 3* vs *Figure 7—figure supplement 4*). We were unfortunately unable to characterize a

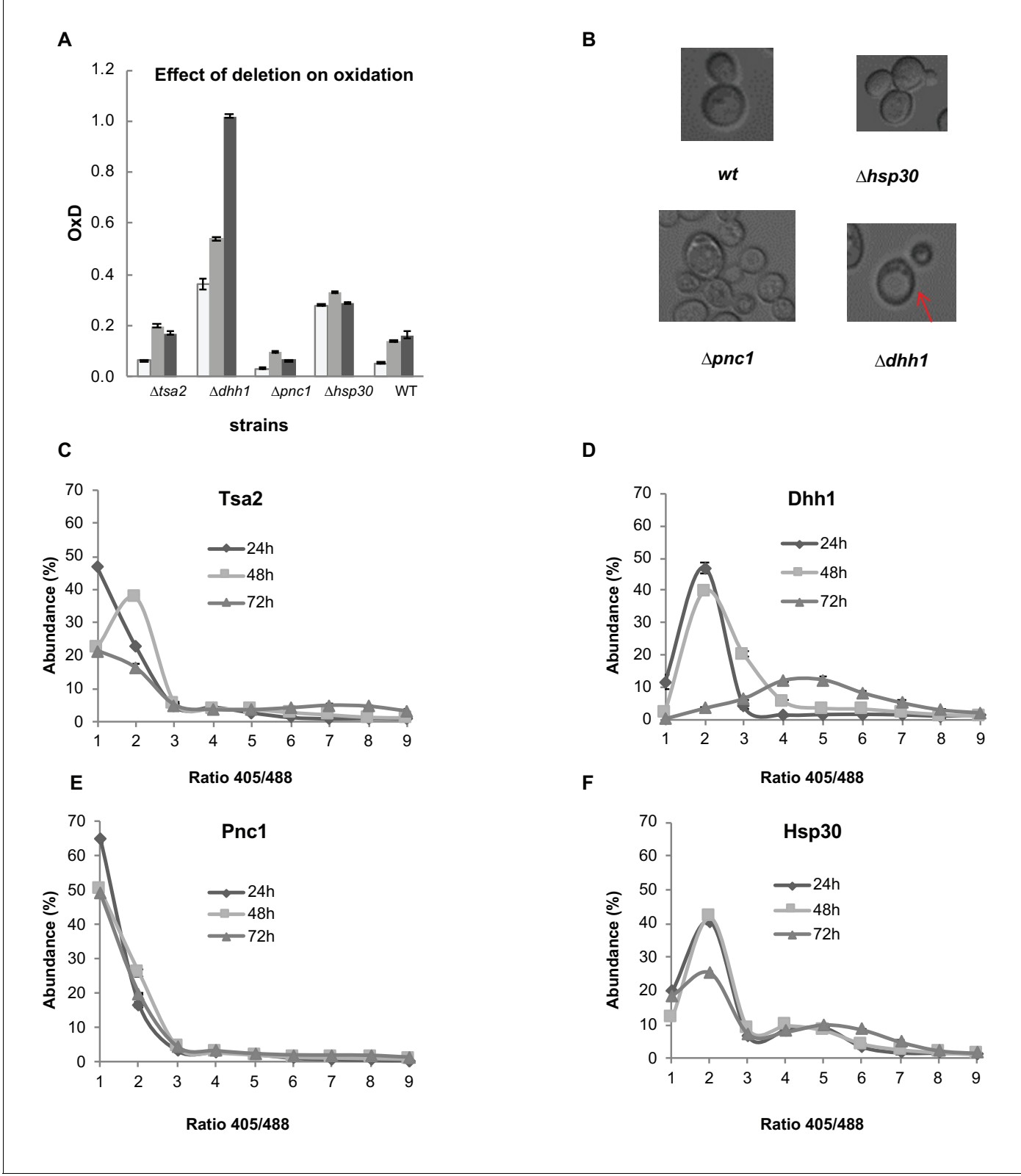

**Figure 7.** Analysis of differentially expressed proteins between the reduced and oxidized subpopulations. (A) Oxidation levels in deletion strains of significantly changed proteins (Δtsa2, Δdhh1, Δpnc1, Δhsp30, and wild type control) at different ages (24, 48, and 72 hr). (B) Differences in cell growth between deletion strains. Δdhh1 has clearly enlarged vacuoles as compared to other strains (emphasized), while Δpnc1 is uniquely small in comparison with the wild type. Δhsp30 presents largely similar to the wild type, with bimodal variation between different cells. (C–F) Distribution of fluorescence

*Figure 7 continued on next page*

*Figure 7 continued*

intensity ratios obtained at 405 and 488 nm of deletion strains. (C) Distribution of Δtsa2, which follows a wild type-like gradual shift towards more highly oxidized ratios. (D) Distribution of Δdhh1, which begins from a slightly higher redox ratio and undergoes a collapse at 72 hr. (E) Distribution of Δpnc1, a protein upregulated in the oxidized subpopulation, which remains highly reduced as compared to the wild type. (F) Bi-modal distribution of Δhsp30, with little variation over time.

DOI: https://doi.org/10.7554/eLife.37623.015

The following figure supplements are available for figure 7:

**Figure supplement 1.** (A–C) Association maps of proteins associated with Dhh1, Pnc1, and Hsp30 (respectively), as identified in the mass spectrometry screen of differentially expressed proteins between the reduced (blue) and oxidized (red) subpopulations.

DOI: https://doi.org/10.7554/eLife.37623.016

**Figure supplement 2.** Dhh1 deletion decreases cell growth.

DOI: https://doi.org/10.7554/eLife.37623.017

**Figure supplement 3.** Oxidation in mitochondria detected by mitochondrial sensor Su9 expressed in wild type, Hsp30 and Pnc1 knockout strains.

DOI: https://doi.org/10.7554/eLife.37623.018

**Figure supplement 4.** Oxidation in peroxisome detected by peroxisomal sensor SKL-roGFP expressed in wild type, dhh1, hsp30 and pnc1 knockout strains.

DOI: https://doi.org/10.7554/eLife.37623.019

crosstalk between the mitochondrial and cytosolic oxidation in the Δ*dhh1* strains due to the low fluorescent signal of the mitochondrial Grx1-roGFP2-Su9 in these strains, possibly due to its unique morphological changes and impairments. In the peroxisome, however, Dhh1 deletion shifted the oxidation toward higher, more oxidized 405/488 nm ratios (*Figure 7—figure supplement 4C*).

Together, these results point towards the potential different roles some of the identified proteins in our screen may contribute to glutathione-dependent redox homeostasis, suggesting distinct modes of altering cellular redox state. More specifically, we have identified three proteins that, when knocked out, produce unique oxidation distribution patterns as compared to the wild type. Further investigation regarding their GSSG/GSH levels, particularly in Δ*dhh1* with its enlarged vacuoles (*Morgan et al., 2013*), would provide greater insight into the mechanism of oxidation/reduction within each strain.

## A differential transcriptome profile of the reduced and oxidized subpopulations

To examine coupling between protein and transcript abundance, we conducted a transcriptomic analysis of three biological replicates of the isolated reduced and oxidized subpopulations at 48 and 72 hr. We identified 4949 genes in the reduced sample and 5027 genes in the oxidized samples (*Supplementary file 6*, *7*).

As expected from the proteomic analysis, global changes in the transcriptome were redox- rather than age-dependent (*Figure 8A,B*, and *Figure 7—figure supplement 4*). Annotation analysis of the differentially expressed transcripts (defined by an at least two-fold change with FDR < 0.05) (*Supplementary files 8–11*) showed relatively similar functional distributions across transcripts from samples of the same age, which correlates well with the annotation of the abundant proteins in these samples. Specifically, the reduced cells had upregulated carbon metabolism and TCA cycle activity. Interestingly, the transcriptomic analysis showed that expression of genes involved in lipid biosynthesis and

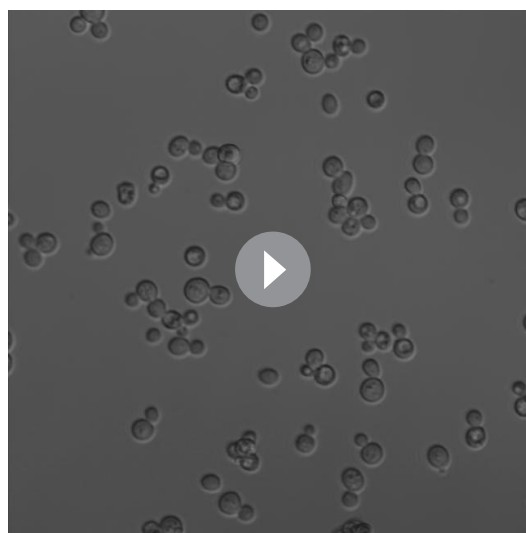

**Video 1.** Growth and replication of the wild type strain. The wild type cells were grown at 30°C on synthetic media for six hours and images were taken every five minutes using time-lapse confocal microscope as described in the Materials and methods part.

DOI: https://doi.org/10.7554/eLife.37623.020

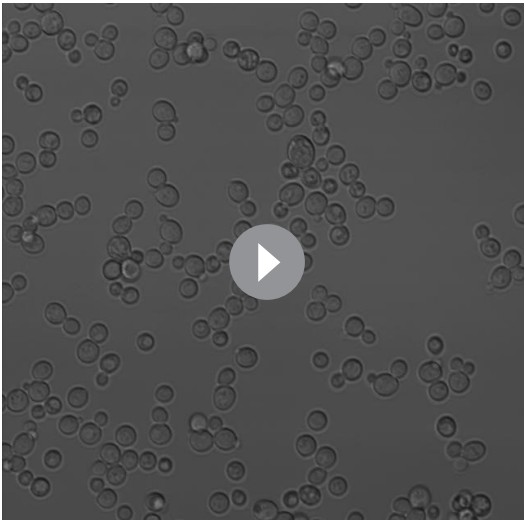

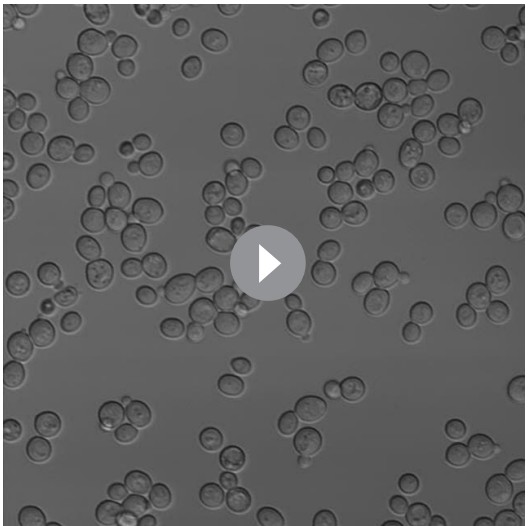

**Video 2.** Growth and replication of the pnc1 knockout strain. The wild type cells were grown at 30°C on synthetic media for six hours and imageswere taken every five minutes using time-lapse confocal microscope as described in the Materials and methods part.
DOI: https://doi.org/10.7554/eLife.37623.021

**Video 3.** Growth and replication of the hsp30 knockout strain. The wild type cells were grown at 30°C on synthetic media for six hours and imageswere taken every five minutes using time-lapse confocal microscope as described in the Materials and methods part.
DOI: https://doi.org/10.7554/eLife.37623.022

peroxisomal function were upregulated. This function was missing from the proteomic analysis, most probably due to low total abundance of peroxisomal proteins (~1%).

In examining the general correlation between the transcriptomic and proteomic analysis at 48 and 72 hr (*Figure 8C*, *Figure 8—figure supplement 1*), we find that while there is a global correlation in the trend (*Figure 8C*, *Figure 8—figure supplement 1*), there are multiple examples of uncoupling in significantly differentially regulated protein and mRNA expression (*Figure 8C*, *Figure 8—figure supplement 1*, dark red and purple bold circles). While other studies have shown that protein expression typically corresponds with mRNA expression, expression of transcripts involved in regulatory roles (*Marguerat et al., 2012*) and aging (*Janssens et al., 2015*; *Wei et al., 2015*) have been found to partially decouple.

One of the intriguing examples is Hsp30 and its homologue, Yro2. Although proteins of these genes were upregulated in the reduced cells, their transcripts were shown to be upregulated in the oxidized cells. We may speculate that cellular oxidation might have a differential impact on the half-life time of Hsp30 mRNA and protein molecules, that is, enhancing Hsp30 protein degradation upon elevated oxidation, maintaining high levels of its transcripts over protein molecules. However, future investigations should be done in order to understand the origin of this phenomenon.

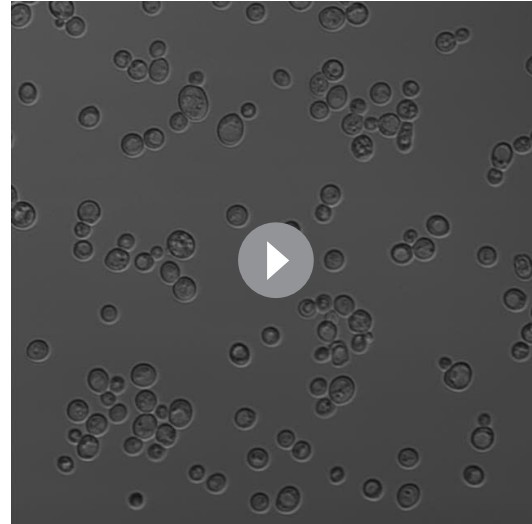

**Video 4.** Growth and replication of the dhh1 knockout strain. The wild type cells were grown at 30°C on synthetic media for six hours and imageswere taken every five minutes using time-lapse confocal microscope as described in the Materials and methods part.
DOI: https://doi.org/10.7554/eLife.37623.023

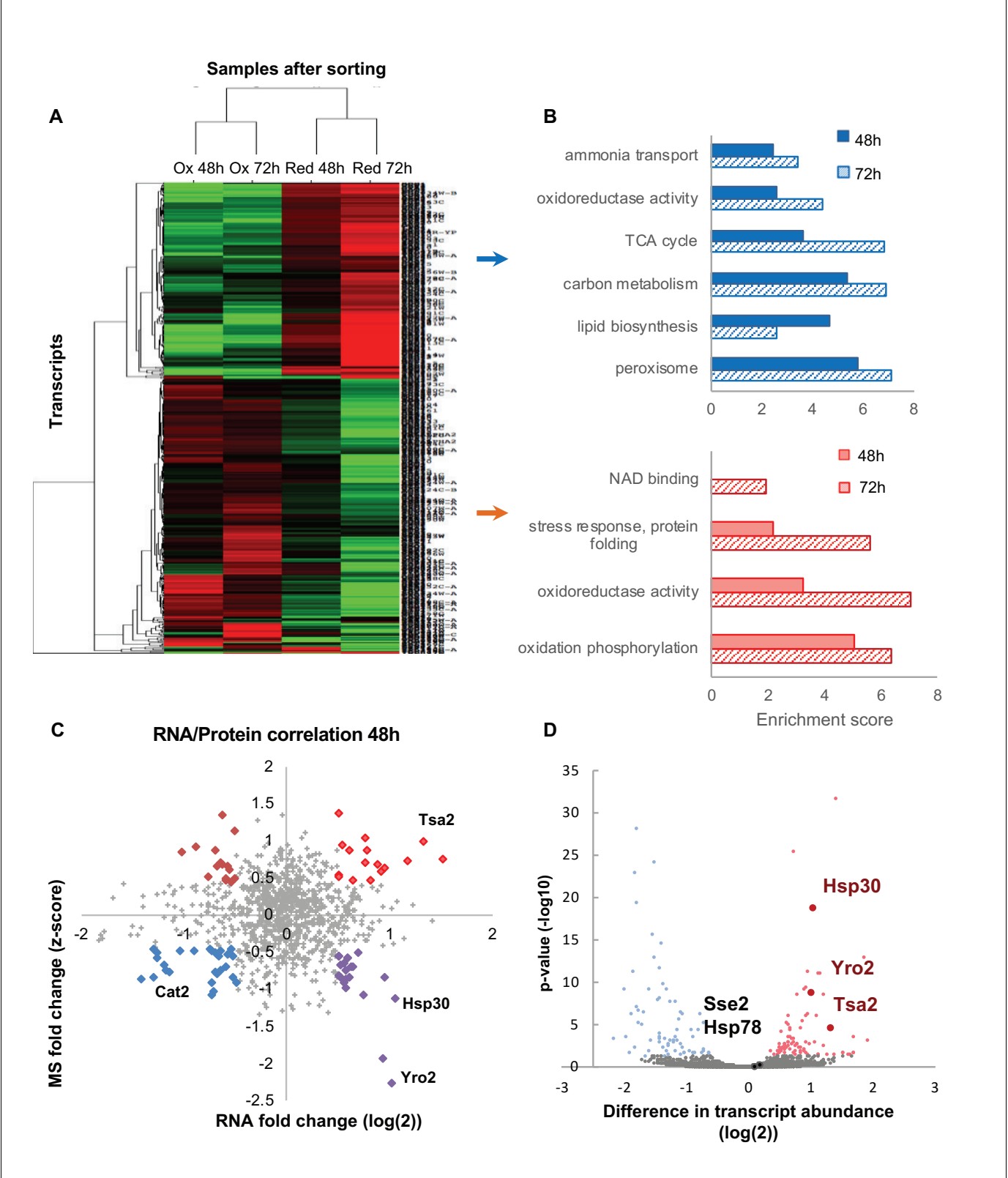

**Figure 8.** Differentially expressed transcripts between the reduced and oxidized subpopulations. (A) Hierarchical clustering of the median expression values of all differentially expressed genes (FDR < 0.05) identified in the post-sorting cells harvested at different time points (48 and 72 hr); downregulated transcripts are in green, up-regulated are in red. Each row was normalized by its median and the log2 value was taken for visualization purposes. The data was clustered using a centered correlation similarity metric. (B) Functional enrichment analysis of the differentially expressed

*Figure 8 continued on next page*

*Figure 8 continued*

transcripts in the reduced (blue) and oxidized (red) subpopulations harvested after 48 hr (solid bars) and 72 hr (solid fill and slashes, respectively). (C) Correlation plot between mRNA and protein expression at 48 hr. Significantly differentially regulated proteins are emphasized in bold and color according to their coupling status (bright red – coupled upregulation in oxidized subpopulation, dark red – uncoupled protein upregulation in oxidized subpopulation, blue – coupled upregulation in reduced subpopulation, purple – uncoupled protein upregulation in reduced subpopulation). (D) The mean value of normalized counts (log2) is plotted for each gene. Each point is colored according to the adjusted value of the differential expression analysis (Materials amd methods). Hsp30, Yro2 and Tsa2 were significantly upregulated in the oxidized cells.

DOI: https://doi.org/10.7554/eLife.37623.024

The following figure supplements are available for figure 8:

**Figure supplement 1.** Hierarchical clustering of expression values of all differentially expressed genes (FDR < 0.05) identified in the biological replicates of the post-sorting cells harvested at 48 and 72 hr.

DOI: https://doi.org/10.7554/eLife.37623.025

**Figure supplement 2.** Correlation plot between mRNA and protein expression at 72 hr.

DOI: https://doi.org/10.7554/eLife.37623.026

As shown previously, Hsp30 is a potential 'redox keeper'; its deletion leads to a bimodal redox distribution in cells, correlating with high protein levels in the reduced subpopulation. To further investigate the impact of Hsp30 and its associated proteins, Yro2, Hsp78 and Sse2 (*Figure 8D*), we examined oxidation levels of these protein knockout strains (*Figure 7—figure supplement 1*, *Supplementary file 12*). While Hsp30 had a significant impact on the redox status of cells, as we have already shown, its orthologue Yro2 did not. In addition, chaperones Hsp78 and Sse2, which showed no significant redox-dependent change in their transcript levels, had no influence on the cellular oxidation.

## The transition from reduced to oxidized cellular state is a threshold-based event

To better understand the switch from reduced to oxidized, we tracked redox changes within individual cells in an attempt to identify how cells transition between these two distinct states.

Using confocal microscopy, we monitored changes in the degree of oxidation (normalized to the cell surface area, henceforth referred to as OxD*, as described in the Materials and methods section) of a specifically chosen cell for 12 hr (*Figure 9A*). To define the OxD* values we measured the fluorescence of fully reduced and fully oxidized cells at 405 and 488 nm (*Figure 9B*), as previously described. These measurements were conducted using diluted, unsorted populations of young cells during the early log phase, during which some cells had either lost their plasmid expression or were otherwise Grx1-roGFP2-negative throughout the 12 hr of growth as measured using confocal microscopy. Nonetheless, we successfully monitored specific OxD* changes in over 30 unique cells across 12 hr (*Figure 9C–F*), and clustered their oxidation trajectories during this time. Interestingly, a subset of these cells maintained their newly reduced state for over 12 hr (the length of the measurement), indicating a strong mechanism for maintaining a reduced environment within the cell (*Figure 9C*). An additional portion of these cells lost their GFP signal without an increase in the oxidation-related fluorescence profile. The majority of the reduced cells were able to 'self-correct' their oxidative status for relatively prolonged periods (*Figure 9D*) or for a shorter time (*Figure 9E*) before undergoing oxidation. Surprisingly, this self-correction process existed only below a clear oxidation threshold, at an OxD* of approximately 0.7 (*Figure 9C–E*). Cells were able to reach OxD*s as high as 0.65 before returning to more reduced values as low as 0.4, at which state they could remain for several hours before crossing the threshold and undergoing rapid oxidation. However, we observed that once cells crossed the oxidation threshold, they were unable to correct their oxidative status and maintained high OxD* values until their eventual cell death (*Figure 9A*). Meanwhile, a cluster of newly born daughter cells began life at different time points from a wide range of OxD*s, undergoing a simple oxidation process over time with little deviation or correction (*Figure 9F*).

Previous research has indicated that in cases of old mother cells with upwards of 10 divisions, the mother cell's age has an impact on its daughters (*Kennedy et al., 1994*), leading to reduced lifespans in the new daughter cells. To test the suggestion that some division-dependent aging factor was somehow capable of being transferred to the daughter cells, we looked at cases of OxD* changes in budding mother cells. We measured the mother cells OxD* during and immediately after

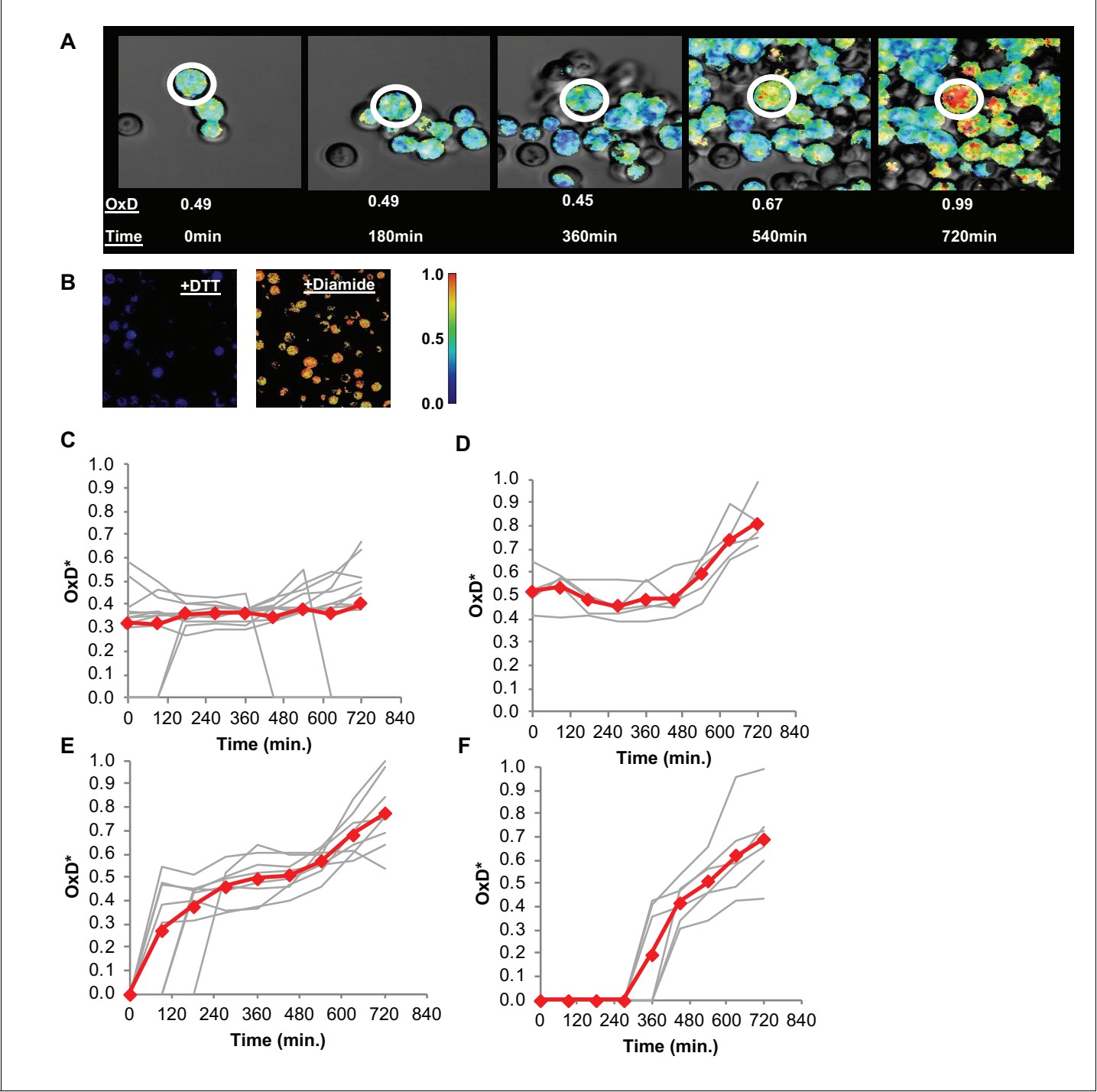

**Figure 9.** Changes in oxidative status of individual cells over time under confocal microscopy. (**A**) Oxidation over time of a single cell, from reduced to highly oxidized, imaged using confocal microscopy. (**B**) Samples treated with 40 mM DTT and 8 mM Diamide for 15 min and imaged using confocal microscopy. (**C–F**) Oxidation of thirty unique single cells over 12 hr, clustered by trajectory similarity (labeled in red).

DOI: https://doi.org/10.7554/eLife.37623.027

division, as well as an hour after division. These cells were chosen on the basis of their clear Grx1-roGFP2 signals, as well as the ability to track changes over a long period of time without overlap or extreme movement. Remarkably, we observed that mother and daughter cells had identical OxD* values throughout division and immediately following separation, regardless of the length of division

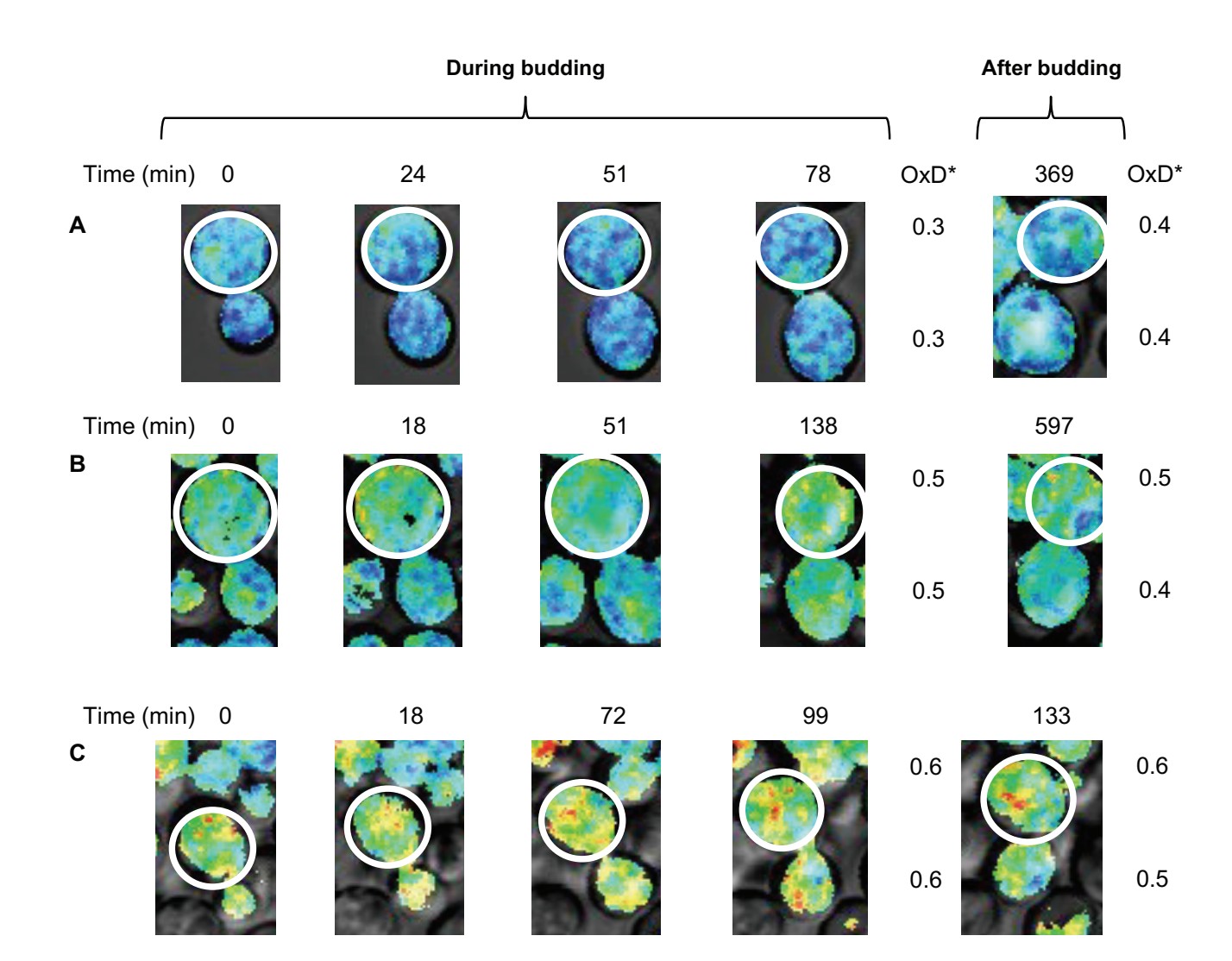

**Figure 10.** Changes in oxidative status between mother and daughter cells over time under confocal microscopy. (A–C) Oxidation levels over time in mother and respective daughter cells during and after budding, showing the shared oxidative status until separation.
DOI: https://doi.org/10.7554/eLife.37623.028

or the mother's initial OxD* (*Figure 10A–C*), thus demonstrating that daughter cells inherited even a relatively highly oxidized environment during budding. Moreover, some daughter cells were able to self-correct their inherited OxD* levels once the budding process was completed and were able to reach a more reduced state within less than an hour, even as the mothers crossed the oxidation threshold or maintained their original OxD* (*Figure 10C*). These findings could suggest that the influence a mother cell has on her daughters may play a more central role in the oxidative stress of the daughter cell in early life.

## Discussion

### Dissection of redox-dependent heterogeneity

Phenotypic heterogeneity within genetically homogenous cell populations is considered one of the strategies for dealing with fluctuating environments and stress conditions (*Elowitz et al., 2002*; *Maamar et al., 2007*; *Avery, 2006*; *Bishop et al., 2007*). Studies in bacteria and yeast have

revealed that cell-to-cell variation is encoded in the stochasticity of gene and protein expression in a normal or changing environment. Diverse mechanisms have been suggested, ranging from promoter activation and deactivation, the rate of mRNA and protein production, protein and gene modifications, and even metabolic exchange between cells (*Campbell et al., 2016*). The typical readout of these processes is diversity in protein expression leading to differential growth under standard or stress conditions.

Here, we identified a new type of glutathione-dependent heterogeneity, originating in the redox status of cells, by focusing on cytoplasmic oxidation alongside chronological and replicative aging. Using a self-established, non-laborious methodology to quantify and isolate cells with different oxidation levels, we suggest that the redox status (up to 72 hr growth in standard medium) is a contributing factor in cellular growth and division properties. Clearly, other parameters, such as energy and glucose levels, will affect age-dependent oxidation. An interesting example of this may be seen in Knieß and Mayer (*Knieß and Mayer, 2016*), in which they find a distinction between fermenting and respiratory conditions on glutathione reduction. Here, they showed an increase in reducing glutathione potential during fermentation, while respiratory cultures have a higher oxidation during replicative aging. This distinction is interesting in comparison with our results regarding replicative aging under aerobic conditions, opening the door to future exploration. Nevertheless, on the basis of our monitoring the time-line of cellular oxidation in aging cells in culture, it is tempting to speculate that the transition from reduced to oxidized status is a threshold-based phenomenon that leads to the emergence of at least two distinct cell subpopulations with different growth potential and biochemical profiles.

Our detection of the redox status of aging yeast cells was based on the cytosolic redox-dependent fluorescent probe, Grx1-roGFP2, which depends on GSH/GSSG levels. The roGFP probes are well-established indicators of in-vivo redox potentials in bacteria (*van der Heijden and Finlay, 2015*), *C. elegans* (*Braeckman et al., 2016*), yeast (*Morgan et al., 2011*), plants (*Meyer et al., 2007*), and mammals (*Dooley et al., 2004*), monitoring differences in oxidative stress under a range of diverse conditions. Measurements of roGFP fluorescence are generally based on microscopy imaging, quantifying the OxD values for each independently or, conversely, deriving an average OxD value for the entire population of suspended cells. While the first approach is able to evaluate cell-to-cell variability, it is time-consuming and arduous. In the latter approach, direct fluorescence measurement of a population is fast but not sensitive enough to detect natural variations within a defined population. Our utilization of FACS combines the advantages of fluorescence measurement at the level of the individual cell with a high throughput ability to isolate living cells according to their redox-related properties. Due to the robustness of the Grx1-roGFP2 sensor, the parameters used to define the oxidation gates to quantify the redox levels in the cytosol, mitochondria and peroxisome were very similar. The discrimination between reduced and oxidized cells was clear-cut, enabling rapid and convenient measurements of biologically different samples collected at different days and conditions. These oxidation gates can be applied to any population of yeast cells expressing the roGFP2 variants to define the contribution of a specific gene or condition on cytosolic redox status or in organelles. Measurements of this sort may thus be conducted on a larger scale using knockout-library arrays, ultimately scanning a wide range of mutations and their redox impact across several days. This may contribute to identifying key proteins serving as regulators of redox-dependent heterogeneity or as 'redox switches', defining the ability to respond to various forms of oxidative and environmental stress as well. More generally, this approach is not restricted to yeast and may be used in any cell type - from bacteria to mammals.

## Identification of potential positive and negative glutathione-dependent modulators of redox homeostasis: Hsp30, Dhh1, and Pnc1

In this study, we identified three proteins that had a significant impact on the redox status of yeast at early and late stages of chronological aging: two negative regulators, Hsp30 and Dhh1 that maintain a reducing environment, and one negative regulator, Pnc1, deletion of which decreases cellular oxidation.

Hsp30 is a plasma membrane heat shock protein which is induced by different stress conditions, including heat shock, exposure to ethanol, glucose limitation, oxidation and entry into the stationary phase (*Piper et al., 1994*; *Seymour and Piper, 1999*; *Thakur and Chakrabarti, 2010*; *Kim et al., 2006*) During heat shock, deletion of Hsp30 leads to inhibition of the Pma1H+ -ATPase, preserving

cellular ATP levels (*Panaretou and Piper, 1992*). Moreover, deletion of Hsp30 is correlated with a decrease in superoxide dismutase (*sod1*) expression (*Thakur and Chakrabarti, 2010*), which might correspond with elevated cellular oxidation. Interestingly, our transcriptomic and proteomic analysis revealed an opposite regulation profile of Hsp30, pointing to an upregulation of gene expression under an oxidizing environment after 48 and 72 hr of growth, alongside downregulation at the protein level at the same time. Moreover, deletion of Hsp30 led to a significant increase in cellular oxidation, suggesting that indeed high Hsp30 protein levels keep the cells reduced. The proteomics analysis showed that upregulation of Hsp30 in the reduced cells was coupled with its native partners, all of which were upregulated in the reduced cells (*Figure 6C*: Dhh1, catalase Ctt1, chaperones Hsp78 and Sse2, Hxt5, and Yro2). Interestingly, an orthologue of Hsp30, Yro2, had a similar protein-transcript anti-correlation; however, deletion of Yro2 had a lesser impact on the redox status than Hsp30.

An antagonistic abundance in the RNA and protein levels suggests a potential impairment of Hsp30 protein turnover in oxidized cells. Most probably, gene expression is induced upon oxidation but protein translation is not efficient enough to sustain levels of Hsp30 that would maintain the reducing environment. This correlates with the downregulation of protein biogenesis and energy production, along with increased proteasome activity in the oxidized cells. Thus, enhanced protein turnover (either due to low translation rates or high proteolysis rates) may push the transcriptional machinery to produce high levels of Hsp30 mRNA without reaching critical protein concentration that serves as a 'redox keeper' of yeast cells.

Furthermore, analysis of the cellular oxidation (405/488 nm fluorescence intensities) pointed to a unique bi-modular distribution of distinct reduced and oxidized populations of cells that are preserved during chronological aging. This suggests that Hsp30 enhances redox heterogeneity, and represents a fascinating role for Hsp30 that may be investigated further in the future.

In addition to Hsp30, we identified another potential 'redox keeper', a highly conserved RNA DEAD-box helicase Dhh1, whose protein levels are upregulated in the reduced cells and whose deletion leads to elevated oxidation from early stages of cell growth. Dhh1 is a stress granule-associated helicase, involved in mRNA metabolism (*Buchan et al., 2010*; *Swisher and Parker, 2010*). In contrast to Hsp30, deletion of the *dhh1* gene led to a global cellular collapse, altering growth, morphology, and possibly inducing high rates of apoptosis. Therefore, changes in cellular oxidation might be a secondary effect accompanying growth arrest. Interestingly, other stress granule-associated proteins were upregulated in the reduced cells, including mRNA cleavage factor Hrp1, the translation initiation factor eIF-4B, Tif3, 5′−3′ exonuclease, Xrn1, and the eIF3b subunit of the eIF3 translation initiation factor, Prt. This might suggest that downregulation of these proteins leads to a collapse in redox status and migration of these proteins to p-bodies for mRNA decay and transcription inhibition. Interestingly, transcripts of all these proteins, except Hrp1, did not show significant change, suggesting that protein turnover or translation rate might play a significant role in the redox homeostasis, rather than gene expression.

In contrast to the positive effect of Dhh1 and Hsp30, the nicotinamidase Pnc1 was found to be upregulated in the oxidized cells. The null mutant had lower OxD levels than the wild type, suggesting that Pnc1 deficiency in cells with normal redox status might lead to reducing stress, while in oxidized cells its upregulation might have a different effect. Previous studies have shown that deletion of Pnc1 increases nicotinamide levels, which inhibit the histone deacetylase Sir2 and lead to apoptosis or a decrease in life span (*Anderson et al., 2003*; *Belenky et al., 2007*; *Gallo et al., 2004*). Thus, we propose that Pnc1 might serve as a link between increased reduced stress, nicotinase accumulation, and a decrease in cell viability.

## Threshold-based mechanism and mother-daughter inheritance underline cellular oxidation

Until now it was believed that most oxidative stress comes from age-dependent damage or genetic alterations. However, an intriguing observation of our study is that there were cells with increased OxD values but no division scars in a genetically identical population. To understand the source of oxidation of these 'newly born' oxidized cells, we decided to monitor changes in the cellular oxidation utilizing real-time imaging over 12 hr. When examining the distribution of oxidation levels across 12 hr, as expected, we found a clear decrease in the number of reduced cells over time, with the peak shifting towards oxidized, Grx1-roGFP2-negative and damaged cells. Notably, the redox status

of cells fluctuated with time, suggesting the activity of a self-repair mechanism until a specific threshold is reached, approximately 70% of maximal oxidation; above this threshold, cells remain permanently oxidized. Analysis of the daughter cells revealed that their oxidation level was identical to that of their mother cells, lasting for several hours 'after birth'. Since the oxidized state received by the daughter cells from the mother is maintained for several hours, and thus exceeds their duplication time, we suspect it is a regulated mechanism and not simple diffusion of the Grx1-roGFP2 sensor with predefined oxidation.

In the future, it would be interesting to characterize modulators of this intrinsic property of cells to maintain their redox status until the observed threshold, as well as to bequeath its redox profile to the daughter cells. For this, advanced technology should be developed enabling sorting cells based on their redox profile monitored during defined period of time. We believe that the methodology described here can be a basis for development of more advanced technology in the future.

## Materials and methods

### Yeast strains, growth conditions and Grx1-roGFP2 probes

The *S. cerevisiae* strains used in this study was the haploid wild-type (BY4741; *MATα, leu2Δ3 his3Δ1 met15Δ0 ura3-0*),as well as BY4741 deletion strains made using the pFA6 KO plasmid (*Longtine et al., 1998*) provided by the Schuldiner laboratory. Additional strains with Mother Enrichment Program genetic modifications (*Lindstrom and Gottschling, 2009*) were provided by the Ravid laboratory. The strains were transformed with Grx1-roGFP2, Grx1-Su9-roGFP2, and Grx1-roGFP2-SKL probes (*Elbaz-Alon et al., 2014*) (kindly provided by Bruce Morgan and Maya Schuldiner, respectively) and grown overnight from plate on 2–4 ml minimal modified casein amino acid (described below) under aerobic conditions, supplemented with amino acids corresponding to plasmid selection. Cultures were then diluted to an $OD_{600}$ of approximately 0.25 on 4–10 ml to ensure fresh growth and brought to their logarithmic phase of $OD_{600}$ of approximately 0.75, which marked 'time zero'. Samples were grown at 30°C under constant agitation. Transformations were refreshed every 2–3 weeks in an adapted version of the protocol from (*Morgan et al., 2011*).

### Modified casein amino acid medium

0.017% yeast nitrogen base (w/o amino acids and ammonium sulfate), 0.5% ammonium sulfate, 2% glucose, 0.2% casamino acid mix, 0.000004% Trp, 0.000005% Thr (amino acids to excess; subject to minor variation), 1% adenine in DDW.

### Calculation of OxD

OxD (the degree of oxidation) was calculated directly from six measured intensities:

$$OxD_{roGFP2} = \frac{I405 * I488_{red} - I405_{red} * I488}{I405 * I488_{red} - I405 * I488_{ox} + I405_{ox} * I488 - I405_{red}I488}$$

This same equation can be rewritten using direct wavelength ratios:

$$OxD_{roGFP2} = \frac{R - R_{red}}{(R_{ox} - R) + (R - R_{red})}$$

Normalization was conducted using the fluorescence intensities of the oxidized/reduced samples measured at 24 hr for all time points.

Calculations of OxD under confocal microscopy was normalized according to surface area rather than total fluorescence. This is referred to as OxD*.

$$OxD^* = \frac{I_{mean\ of\ area\ of\ cell} - I_{red}}{I_{ox} - I_{red}}$$

### Monitoring yeast growth

Growth curves were conducted on sorted yeast samples using a TECAN Infinite 200 PRO series, using an adapted version of the protocol from (*Morgan et al., 2011*). Measurements were initiated using 4.5 million cells sorted according to their redox status. Yeast incubation and $OD_{600}$

measurements, as well as minimal doubling times, were analyzed using MDTCalc. Minimum doubling time was assessed by first calculating the log10 value of each point during logarithmic growth, and then measuring the slope between each set of consecutive points. The value of this slope was then inverted (representing the minimal time it took the sample point to 'double'), providing the minimal doubling time value.

## Flow cytometry analysis of redox ratios and sorting

Yeast cells transformed with Grx1-roGFP2 probes and treated with 1:250 (1 µl) propidium iodide Sigma p4170 for 15 min (adapted from the protocol described by Ocampo at al [*Ocampo and Barrientos, 2011*]) were measured during their extended stationary phase every ~24 hr using no more than 1.0 $OD_{600}$ (10 µl), suspended in 240 µl phosphate buffered saline (PBS x 1). Total oxidation and reduction of the cultures were determined through addition of 10 µl 1M dithiothreitol (DTT) (40 mM) and 10 µl 0.2M diamide (8 mM) to 230 µl PBS x 1. Flow cytometry analysis was performed using a sorting-equipped FACS Aria III flow cytometer, with 405, 488 and 531 nm lasers (BD Biosciences, San Jose, CA) and the flow cytometry data was analyzed using FACSDiva software (BD Biosciences). Voltage settings for the SSC, FSC, 405 nm and GFP channels were kept constant for all experiments. For maximal discrimination between reduced and oxidized Grx1-roGFP2, we used 2D dot plots with linear rather than log scales. Measurements were taken at 405 and 488 nm for OxD calculations and redox ratio, while dead cells were gated out using propidium iodide labeling (excitation at 531 nm and emission detected by a 660 nm filter). Each analyzed population had a sample size of 10,000 standard cells.

## Sample preparation for mass spectrometry

Sorting was based on the redox ratio, using 9 million cells suspended in PBS. Cells were then lysed using 300 µl 0.2M NaOH and resuspended using a lysis buffer (100 mM DTT [15.425 mg DTT], 100 µl 1M Tris HCl pH 7.5, 100 µl SDS 20%, complete with DDW to 1 ml). Samples were then diluted using 400 µl of a urea buffer (8M urea in 0.1M Tris HCl pH 8.5), loaded onto a filter, and centrifuged for 10 min at 12,000 g following the standard FASP protocol (*Wiśniewski et al., 2009*). This process was repeated three times, with flow-through discarded, after which samples were incubated in the dark for 60 min with 0.5M iodoacetamide and urea buffer (final iodoacetamide concentration of 0.05M), under constant agitation (350 rpm, 25°C). Samples were then washed again three times with urea buffer and twice with digestion buffer (10% ACN, 25 mM Tris HCl pH 8.5), then centrifuged at 12,000 g for 8 min. Filters were transferred to a new collection tube, suspended in 300 µl of digestion buffer with 1 µl of trypsin (Promega), mixed for 1 min at 600 rpm, and left overnight at 350 rmp, 37°. Following digestion, samples were centrifuged for 10 min at 12,000 g.

The peptide concentration was determined, after which the peptides were loaded onto stage tips in equal amounts. Stage tips were activated using 100 µl MS-grade methanol (100% MeOH) and centrifuged for 2 min at 2,000 g, after which they were cleaned with 100 µl of elution buffer (80% ACN, 0.1% formic acid) and centrifuged again for 2 min at 2,000 g. The stage tips were returned to their hydrophilic state by suspension in 100 µl of buffer A (0.1% HPLC-grade TFA) and centrifugation at 2,000 g for 2 min, repeated once. 10–30 µg of protein was then loaded per stage tip (as per protein preparation above), and centrifuged at 1,000 g for 2 min. Proteins were then washed twice with 100 µl buffer A at 1,000 g for 2 min and transferred to a new collection tube. Peptides were eluted using 60 µl buffer B (80% ACN, 0.1% HPLC-grade TFA) centrifuged at 250 g for 2 min, and another 30 µl buffer B centrifuged at 250 g for 2 min. Samples were then dried using a SpeedVac for 18 min at 1,300 rpm at 35°, after which they were dissolved in 6–12 µl of buffer A and prepared for tandem mass spectrometry analysis.

## Nano-LC-MS/MS analysis

The peptides were injected into a Nano Trap Column, 100 µm i.d. ×2 cm, packed with Acclaim PepMap100 C18, 5 µm, 100 A° (Thermo Scientific) for 8 min at flow 5 ul/min, and then separated on a C18 reverse-phase column coupled to the Nano electrospray, EASY-spray (PepMap, 75 mm x 50 cm, Thermo Scientific) at flow 300 nl/min using an Dionex Nano-HPLC system (Thermo Scientific) coupled online to Orbitrap Mass spectrometer, Q Exactive Plus (Thermo Scientific). To separate the peptides, the column was applied with a linear gradient with a flow rate of 300 nl/min at 35°C: from

1% to 35% in 100 min, from 35% to 55% in 43 min, from 55% to 90% in 5 min, and held at 90% for an additional 30 min, and then equilibrated at 1% for 20 min (solvent A is 0.1% formic acid, and solvent B is 80% acetonitrile, 0.1% formic acid). The Q Exactive was operated in a data-dependent mode. The survey scan range was set to 200 to 2000 m/z, with a resolution of 70,000 at m/z. Up to the 12 most abundant isotope patterns with a charge of $\geq$2 and less than seven were subjected to higher-energy collisional dissociation with a normalized collision energy of 28, an isolation window of 1.5 m/z, and a resolution of 17,500 at m/z. To limit repeated sequencing, dynamic exclusion of sequenced peptides was set to 60 s. Thresholds for ion injection time and ion target value were set to 70 ms and $3 \times 10^6$ for the survey scans and to 70 ms and $10^5$ for the MS/MS scans. Only ions with 'peptide preferable' profile were analyzed for MS/MS. Data was acquired using Xcalibur software (Thermo Scientific). Column wash with 80% ACN for 40 min was carried out between each sample run to avoid potential carryover of the peptides.

## Data analysis and statistics of the proteomic data

For protein identification and quantification, we used MaxQuant software (*Cox and Mann, 2008*), version 1.5.3.30. We used Andromeda search incorporated into MaxQuant to search for MS/MS spectra against the UniProtKB database of Saccharomyces cerevisiae proteome, (Uniprot release, Aug 2016). The identification allowed two missed cleavages. Enzyme specificity was set to trypsin, allowing N-terminal to proline cleavage and up to two miscleavages. Peptides had to have a minimum length of seven amino acids to be considered for identification. Carbamidomethylation was set as a fixed modification, and methionine oxidation was set as a variable modification. A false discovery rate (FDR) of 0.05 was applied at the peptide and protein levels. An initial precursor mass deviation of up to 4.5 ppm and fragment mass deviation up to 20 ppm were allowed. Only proteins identified by more than two peptides were considered. To quantify changes in protein expression we used the label-free quantification (LFQ) using the MaxQuant default parameters (*Cox and Mann, 2008*). For statistical and bioinformatic analysis, as well as for visualization, we used Perseus software (http://141.61.102.17/perseus_doku/doku.php?id = start). For functional enrichment analysis, the DAVID webserver (*Huang et al., 2009*) was used. The STRING server (http://string-db.org/) (*Szklarczyk et al., 2015*) was used to define protein interaction networks, which were visualized by using Cytoscape software (*Shannon et al., 2003*). Proteomic data was uploaded to the PRIDE database (*Wang et al., 2012*) with the dataset identifier PXD009443.

## RNA purification for the transcriptomic analysis

Cells were incubated with Proteinase K (Epicentre MPRK092) and 1% SDS at 70°C to release the RNA. Cell debris was precipitated by centrifugation in the presence of KOAc precipitation solution. Finally, the RNA was purified from the supernatant using nucleic acid binding plates (UNIFILTER plates, catalog #7700–2810) and was stored with RNAse-inhibitor (Murine #M0314L) at −80°C.

## 3' RNA library preparation

Total RNA (~20 ng per sample) was incubated with oligo-dT RT primers with a 7 bp barcode and a 8 bp UMI (Unique Molecular Identifier) at 72°C for 3 min and transferred immediately to ice. RT reaction was performed with SmartScribe enzyme (TaKaRa Lot# 1604343A) at 42°C for one hour followed by incubation at 70°C for 15 min. Barcoded samples were then pooled and purified using SPRI beads X1.2 (AMPure XP). DNA-RNA molecules were tagmented using Tn5 transposase (loaded with oligo TCGTCGGCAGCGTCAGATGTGTATAAGAGACAG) and 0.2% SDS was used to strip off the Tn5 from the DNA, followed by a SPRI X2 clean up. NGS sequences were added to the tagmented DNA by PCR (KAPA HiFi HotStart ReadyMix 2X (KAPA Biosystems KM2605), 12 cycles). Finally, DNA was purified using X0.65 SPRI beads followed by X0.8 SPRI beads. The library was sequenced using Illumina NextSeq-500 sequencer.

## RNA sequence analysis

Reads were mapped to the yeast genome (sacCer3) using bowtie2 with default parameters (*Langmead et al., 2009*). Duplicated reads were filtered using UMI, to remove PCR bias. To estimate the expression level of each gene we counted the number of reads that mapped to the 3' end of the gene (from 350 bp upstream to 200 bp downstream of TTS). The read counts in each

sample were normalized to PPM (divided by the total number of reads and multiplied by $10^6$). The P-value of the differential expression analysis was obtained as described by Anders and Huber (*Anders and Huber, 2010*), and corrected for multiple hypothesis testing using FDR. Transcriptomic data was uploaded to the GEO database (https://www.ncbi.nlm.nih.gov/geo/query/acc.cgi?acc=GSE112997).

## Imaging of the yeast cells
### Microscopy used for *Figure 2*
Yeast strains were grown on synthetic media without uracil for selection. Imaging was performed using ScanR automated inverted fluorescent microscope system (Olympus). Images of cells were obtained in 384-well plates at 24°C using a 60 × air lens (NA 0.9) with an ORCA-ER charge-coupled device camera (Hamamatsu). Images were acquired in the GFP channel (excitation filter 490/20 nm, emission filter 535/50 nm).

### Confocal microscopy used for *Figures 8–9*
The yeast cells were grown with anti-fluorescent medium, supplemented with casein and amino acids corresponding to plasmid selection and filtered with 0.22 uM filters. Samples were grown overnight on anti-fluorescent medium diluted 1:10 and again 1:3 at $OD_{600}$ of approximately 0.25. 500 µl of sample was then placed on sterile µ-slides (Ibidi, GmbH, Munich, Germany) coated with Concanavalin A (C2272 SIGMA) and incubated several minutes before taking out excess. Cells were observed using time-lapse confocal microscopy (Olympus FV-1200) with 405 and 488 nm lasers. Images were taken every 5–20 min for 6–12 hr while at 30°C and were analyzed using the ImageJ software.

## Seahorse assay
Samples were collected after sorting by centrifugation at 3,700 g for 5 min. The pellet was resuspended in synthetic media without uracil, supplemented with 2% glucose (*Simpkins et al., 2016*), and transferred into an eight welled Seahorse XFp Flux Analyzer Miniplates (Seahorse Bioscience) coated previously with Concanavalin A (Sigma-Aldrich). Each well was loaded with $3*10^6$ cells and miniplates were centrifuged at 50 g for 1 min (*Ghosh et al., 2014*); the plates were then incubated in 30°C for 20 min without $CO_2$. The Seahorse XFp Extracellular Flux Cartridge was incubated with the XF calibrant overnight at 30°C per manufacturers instruction and was used to calibrate the Seahorse XFp Analyzer. The miniplates were then analyzed with measurement of the basal oxygen consumption rates (OCR) and extracellular acidifications rates (ECAR) that correlate to oxidative - phosphorylation and glycolysis, using the following parameters: initial time 12 min, to allow equilibration of the samples, mixing 3 min, measuring 3 min, the temperature was set to 30°C.

## Scar counting and budding
Samples were collected after sorting by centrifuging at 3,700 g for 5 min; the pellet was resuspended with 500 µl of PBS. After transferring liquid to Eppendorfs, samples were centrifuged at max speed for 1 min and the pellet was resuspended with 4% paraformaldehyde and incubated 10 min. Paraformaldehyde was washed with PBS. The pellet was resuspended with 20 µl of PBS and 2 µl of Calcofluor (18909 SIGMA-ALDRICH) and incubated at room temp for 10 min. Samples were washed with PBS before placing on glass slides. Images were taken by confocal microscopy (Olympus FV-1200) and kept at $30^c$ throughout the process. Scars and buds were analyzed using the ImageJ software and manually counted.

## Mother enrichment program system and scar count assessment
Cells were labeled with Sulfo-N-Hydroxysuccinimide-LC Biotin (Sulfo-NHS-LC Biotin) at a concentration of $2 \times 10^4$/mL and stored on glycerol at −80°C until resuspension in filtered minimal modified casein amino acid medium containing 1 µM estradiol. The estradiol is necessary to induce elimination of the daughter cells, according to the Mother Enrichment Program. These samples were incubated for various time periods (0–72 hr) and harvested together. Cells were pelleted by low speed centrifugation, followed by treatment with Texas Red-X conjugate of wheat germ agglutinin (WGA) lectin (W21405, ThermoFisher Scientific) at a concentration of 0.02 mg/mL, which preferentially binds to bud scars (*Chen and Contreras, 2004*). Cells were then analyzed using flow cytometry with

excitation at 405, 488, and 561 nm (as described earlier). Unlike earlier analysis, bud scar assessment was approximated into two groups, 'high scar count' and 'low scar count'. Due to the expanded size of cells following increased division, the range of analyzed cells was enlarged accordingly to more accurately represent the new population. Similarly, detection of the oxidized and reduced subpopulations shifted slightly with the increased size and the corresponding gates were very moderately adjusted.

## Acknowledgements

We are extremely grateful to the director of the confocal unit, Naomi Melamed-Book for assistance and running of the cell imaging assays. We thank Bruce Morgan for providing us with the Grx1-roGFP2 expressing plasmids. We received financial support from the Marie-Curie integration grant (project number: 618806), the Israel Science Foundation (grant number: 1765/13), Human Frontier Science program (CDA00064/2014), the US-Israel Binational Science Foundation (grant number: 2015056), and the Legacy Heritage Biomedical Science Partnership (a program in the Israel Science Foundation; grant number: 1649/16). SI was supported by the Berlin-Hebrew University fellowship.

## Additional information

### Funding

| Funder | Grant reference number | Author |
|---|---|---|
| Hebrew University of Jerusalem | Joint Berlin-Jerusalem postdoc fellowship | Sidra Ilyas |
| Freie Universität Berlin | Joint Berlin-Jerusalem postdoc fellowship | Sidra Ilyas |
| Israel Science Foundation | 1765/13 | Dana Reichmann |
| Human Frontier Science Program | CDA00064/2014 | Dana Reichmann |
| European Commission | | Dana Reichmann |
| United States-Israel Binational Science Foundation | 2015056 | Dana Reichmann |
| Marie Curie Career Integration | 618806 | Dana Reichmann |
| Israel Science Foundation | Legacy Heritage Biomedical Science Partnership, 1649/16 | Dana Reichmann |

The funders had no role in study design, data collection and interpretation, or the decision to submit the work for publication.

### Author contributions

Meytal Radzinski, Data curation, Formal analysis, Validation, Investigation, Visualization, Methodology, Writing—original draft, Writing—review and editing, Designed the research and related experiments, Established the FACS-based redox quantification, Performed distribution data analysis, Performed yeast sorting, Performed proteomic analysis of the sorted yeast, Designed and performed replicative aging experiment; Rosi Fassler, Performed yeast sorting, Performed proteomic analysis of the sorted yeast, Designed and performed yeast respiration experiment, Performed the cell imaging and scar detection; Ohad Yogev, Performed yeast sorting, Performed proteomic analysis of the sorted yeast; William Breuer, Supervision, Methodology, Writing—review and editing, Established the FACS-based redox quantification; Nadav Shai, Resources, Writing—review and editing, Designed the yeast strains expressing the Grx1-roGFP2-SKL sensor; Jenia Gutin, Data curation, Formal analysis, Validation, Visualization, Writing—review and editing, Performed the transcriptomic analysis of sorted cells; Sidra Ilyas, Data curation, Formal analysis, Investigation, Visualization, Designed the research and related experiments, Performed distribution data analysis; Yifat Geffen, Resources, Data curation, Formal analysis, Investigation, Methodology, designed the research and

related experiments, Designed and performed replicative aging experiment; Sabina Tsytkin-Kirschenzweig, Data curation, Supervision, Visualization, Methodology, Writing—review and editing, Designed the research and related experiments, Designed and performed yeast respiration experiment; Yaakov Nahmias, Supervision, Visualization, Methodology, Writing—review and editing, Designed the research and related experiments, Designed and performed yeast respiration experiment; Tommer Ravid, Supervision, Methodology, Writing—review and editing, Designed and performed replicative aging experiment; Nir Friedman, Supervision, Visualization, Methodology, Writing—review and editing, Designed the research and related experiments; Maya Schuldiner, Supervision, Visualization, Methodology, Writing—review and editing, designed the research and related experiments, Designed the yeast strains expressing the Grx1-roGFP2-SKL sensor; Dana Reichmann, Conceptualization, Formal analysis, Supervision, Funding acquisition, Validation, Visualization, Methodology development, Writing—original draft, Writing—review and editing, Designed the research and related experiments, Performed proteomic analysis of the sorted yeast, Performed distribution data analysis

## Author ORCIDs
Nadav Shai (iD) https://orcid.org/0000-0002-2812-3884
Nir Friedman (iD) https://orcid.org/0000-0002-9678-3550
Dana Reichmann (iD) http://orcid.org/0000-0003-0315-5334

## Decision letter and Author response
Decision letter https://doi.org/10.7554/eLife.37623.047
Author response https://doi.org/10.7554/eLife.37623.048

# Additional files

## Supplementary files
• Supplementary file 1. Comparison of OxD values of wild type and knockout strains (related to *Figure 2F*).
DOI: https://doi.org/10.7554/eLife.37623.029

• Supplementary file 2. Protein list.
DOI: https://doi.org/10.7554/eLife.37623.030

• Supplementary file 3. Significant proteins (48 hr).
DOI: https://doi.org/10.7554/eLife.37623.031

• Supplementary file 4. Significant proteins (72 hr).
DOI: https://doi.org/10.7554/eLife.37623.032

• Supplementary file 5. Comparison of wild type and knockout strains OxD values (related to *Figure 6E*).
DOI: https://doi.org/10.7554/eLife.37623.033

• Supplementary file 6. Transcript list (48 hr).
DOI: https://doi.org/10.7554/eLife.37623.034

• Supplementary file 7. Transcript list (72 hr).
DOI: https://doi.org/10.7554/eLife.37623.035

• Supplementary file 8. Transcripts upregulated in red (48 hr).
DOI: https://doi.org/10.7554/eLife.37623.036

• Supplementary file 9. Transcripts upregulated in ox (48 hr).
DOI: https://doi.org/10.7554/eLife.37623.037

• Supplementary file 10. Transcripts upregulated in red (72 hr).
DOI: https://doi.org/10.7554/eLife.37623.038

• Supplementary file 11. Transcripts upregulated in ox (72 hr).
DOI: https://doi.org/10.7554/eLife.37623.039

• Supplementary file 12. Comparison of wild type and knockout strains OxD values (related to *Figure 7*).

DOI: https://doi.org/10.7554/eLife.37623.040
• Transparent reporting form
DOI: https://doi.org/10.7554/eLife.37623.041

## Data availability

All data generated or analyses during this study are included in the manuscript and supporting files. Proteomic data was uploaded to the PRIDE database with the dataset identifier PXD009443. Transcriptomic data was uploaded to the GEO database as described in the manuscript (methods).

The following datasets were generated:

| Author(s) | Year | Dataset title | Dataset URL | Database, license, and accessibility information |
|---|---|---|---|---|
| Meytal Radzinski, Ohad Yogev, Dana Reichmann | 2018 | Proteomic analysis of the natively reduced and oxidized yeast cells | https://www.ebi.ac.uk/pride/archive/projects/PXD009443 | Publicly available at EBI PRIDE (accession no: PXD009443) |
| Reichmann D | 2018 | Transcriptomic data from | https://www.ncbi.nlm.nih.gov/geo/query/acc.cgi?acc=GSE112997 | Publicly available at the NCBI Gene Expression Omnibus (accession no: GSE112997) |

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
