## [Decision Letter]

Thank you for submitting your work entitled "Temporal profiling of redox-dependent heterogeneity in single cells" for consideration by *eLife*. Your article has been evaluated by a Senior Editor and three reviewers, one of whom, Agnieszka Chacinska (Reviewer #1), is a member of our Board of Reviewing Editors. The following individual involved in review of your submission has agreed to reveal their identity: Michel B Toledano (Reviewer #2).

Our decision has been reached after consultation between the reviewers. As you will see from the individual comments, the reviewers appreciated the interesting question and a rich set of high quality data. However, in the current form the manuscript seems to be at the stage that is not sufficiently mature to invite well-defined additions and comments. Summarizing the reviewer’s comments, the paper requires some major shaping based on the three points, to 1) streamline and focus the story, 2) expand some parts experimentally and 3) apply more rigor when drawing the conclusions.

Ad. 1) as suggested in the individual reviews some part may be either omitted or better linked to the rest of the paper, including some additional analyses if required.

Ad. 2) several important issues have been raised. The manuscript definitely requires the clarification in terms of replicative vs. chronological aging, and clarification of the respiratory status of the cells. Furthermore, some of the other aspects should be addressed, e. g. specifying the role of the redox-changing proteins identified in this study. In case, if experimental clarification goes beyond the scope or is technically too challenging, the authors should carefully phrase the conclusions in reference to the point 3).

Given the high potential of the story seen by the reviewers, we would like to encourage the resubmission of the manuscript. To have high chances in *eLife*, the manuscript must be properly revised in terms of all three points mentioned above, including new experimental data. The revised version of the manuscript will likely be evaluated by the same reviewers who are willing to have another look.

*Reviewer #1:*

This is an interesting resource paper that identifies a phenotypic variability in the yeast population in terms of the cellular redox state of glutathione. The authors used an in vivo redox sensor to isolate different populations of cells being with a more oxidized or more reduced status. The populations have been characterized for their fitness and systematically analyzed to identify the differences in the abundance of proteins and in transcriptomics.

The paper is rich in data centered around a really interesting question. It gives a first systemic glance on the molecular changes underlying the phenomenon of variability in redox states of the cells and their dynamics in time and during cell division. The data are slightly descriptive, but this is a resource paper.

The following points should be taken under considerations to revise the work:

- The three proteins identified in the study based on their enrichment in the reduced or oxidized state were functionally examined and concluded to take part in dynamic processes that shape the redox potential of the cell. Hsp30 was in fact implicated earlier. Having established such exciting technology the authors are able to ask more precise questions on which stage these proteins act. Does their absence modify the dynamics of cellular redox changes in the entire population by changing the redox state values of both population groups? Do they affect a bimodal behavior observed in the cell population, i.e. by modifying the number of reduced vs oxidized cells? Does their absence change the ability to enter into the irreversible phase of the oxidized state, or perhaps the daughters of old oxidized mothers are less efficiently recovering into a reduced state?

- There are plenty of examples, in this study and literature, that the transcriptomics and proteomics frequently do not agree perfectly well. It would interesting to undertake more systematic analyses to see the extent of correlations.

- This is somewhat not surprising that the daughter cell inherits the redox status, also the oxidized status, from the mother. An interesting and surprising thing is however that some daughter cells may retain the ability to recover as briefly mentioned at the end of the Results section. This part is somehow slightly underdeveloped in the manuscript. Could the authors develop this part further?

- The authors should avoid bold statements. As already mentioned, the results are largely of correlative nature, and causes and consequences are unclear. For example, I do not see the basis for the following statement: "…we suggest that the redox status, rather than cell age, is a critical determinant of cellular growth and division properties". The text should be carefully evaluated to avoid the statements that do not have a source in literature or in the authors data.

*Reviewer #2:*

In this paper, Reichmann and colleagues measured the redox state of the GSH redox probe Grx1-roGFP2 in the cytosol of yeast cells during chronological aging, and found that the probe oxidized overtime. They could then distinguish two subpopulations, one that remained reduced, but diminished during aging, and the other more oxidized that increased. They sorted these two populations by FACS, and inspected their growth phenotype, their transcriptome and proteome. They identified 4 proteins that are differently expressed in the two subpopulations, and tested their influence on the redox state of roGFP2 by the effect of the corresponding gene deletions. They conclude on the presence of a redox heterogeneity within the same cell population that can be transmitted to daughter cells.

This is an interesting paper that addresses a wild unexplored question of possible redox heterogeneity within a yeast population, and provides an intriguing answer to as the basis of the observed heterogeneity, and possible role of a few selected protein in redox control/aging. The enormous amount of work presented appears very rigorously performed.

There is, however, a concern regarding the interpretation of the data of Figure 4, on which the paper's conclusion is built up. Authors distinguished within a yeast population undergoing chronological aging two subpopulations with a different roGFP2 redox state. Upon inspection of these populations, they found that the reduced one had very few bud scars (< 1), and the oxidized one 3-4.5 more scars: the question is whether the two subpopulations differ by their replicative aging status, with those remaining reduced being newly born daughter cells, and those that oxidize being mother cells, which already underwent a few division, which would be consistent with their growth phenotype, and with the notion that cells oxidize Grx1-roGFP2 during replicative aging, when grown in a respiratory medium (Kniess and Mayer, 2016)? If this the case would collapse the authors' claim that the differences of the two subpopulations are redox, intrinsically, and not age-dependent.

There are a few other points that must be addressed, as described below.

1) Authors must evaluate the possible difference in replicative aging status between the two cells subpopulations: At best, they could sort old and young cells, based on the number of budscars, as done by Kniess and Mayer, and evaluate the redox state of their cytosol during chronological aging, or simultaneously sort cells according to their redox state and budscar status.

2) Table 1 indicate the presence of 30% of reduced cell undergoing budding: It would seem however that chronological aging is the survival of a post-replicative stationary culture: please comment.

3) The mitochondrial status (respiration) should also be established within the two subpopulations.

4) Figure 3C: it is surprising to see that the increase in oxidation during aging gives oxidation values (ratio 405/488) higher than the one measured with diamide (Figure 1D). Please, comment. At best, data should be displayed here using OxD, rather than ratios of fluorescence.

5) The values of OxD measured in the cytosol are much higher than published ones using the same probe (Kojer et al., 2012): 0.3 in Wt cytosol (Figure 2A) vs. 0.1. These values are also different in the different figures, which makes the comparison with mutants difficult. In Figure 6E and 7D values in Wt are < 0.1 and those of mutant similar to Wt in 2A. The total probe oxidation of ddh1D at 96 h is very surprising: are these cells dying? Please comment.

6) The first figures of the paper do not appear really informative and might distract/confuse the reader: Figure 1 displays published data (Kojer et al., 2012). Figure 2 does not provide any data that fit the current story, and use of Glr1 in 3 is not informative because it is known that Glr has a dominant function in reducing glutathione.

*Reviewer #3:*

In their manuscript "Temporal profiling of redox-dependent heterogeneity in single cells" Radzinski et al. present a method to track the redox state of individual cells using a combination of genetically encoded fluorescent redox probes and FACS. Using this methodology, they discover a bi-modal distribution of reduced and oxidized yeast cells in a chronologically aging population. They then proceed to analyze these 2 subpopulations using proteomics and transcriptomics and follow up on some of their findings, identifying Hsp30, Dhh1 and Pnc1 as potential regulators of the cellular redox homeostasis.

The authors present an impressive amount of data and overall, their results suggest that current assumptions about the redox state in aging cells are too simplistic and need to be revised. A model that assumes a simultaneous and uniform decrease in the redox state in all cells of an aging population must be dismissed. And even when considering reproductive aging, the authors could show that daughters can "inherit" the oxidation state of their mothers.

However, some of their data analyses do not seem very productive, in particular some of the STRING-based interaction networks and the functional enrichment. A removal of parts of these analyses would probably not affect their overall conclusions. Should they decide to keep these analyses, they need to justify them and describe these analyses in more detail: What were the criteria chosen by the authors to identify an interaction in the interaction networks? What does it e.g. mean if "Hsp30 is connected to Dhh1"? Is it justified to have proteins that were up-regulated in both reduced and oxidized sub-populations in the very same interaction network? What does it mean, if proteins that "interact" are upregulated under diametrically opposed conditions? How were functions defined for the functional enrichment analyses?

The authors should also submit their proteomics and transcriptomics data to appropriate repositories (e.g. the EMBL PRIDE archive, ArrayExpress, or GEO).

[Editors’ note: what now follows is the decision letter after the authors submitted for further consideration.]

Thank you for resubmitting your work entitled "Temporal profiling of redox-dependent heterogeneity in single cells" for further consideration at *eLife*. Your revised article has been favorably evaluated by Didier Stainier (Senior Editor) and three reviewers, one of whom is a member of our Board of Reviewing Editors.

The reviewers agree that the author did a great job. The reviewers were in fact impressed by the amount and quality of the majority of data. Overall the manuscript is rich in interesting and valuable conclusions. However, there are still some questions raised that require text clarification and discussion. These points are appended below and should be carefully and fully addressed and some shortcomings and limitations should be acknowledged.

We would like to draw your attention to the issue of the oxidation data (the point 2 of reviewer 2). These data should be provided in a full version, and not only as ratios.

*Reviewer #2:*

1) The heterogeneity in probe oxidation the authors saw during chronologic aging might have been the result of differences in replicative aging, and not just of cellular heterogeneity; in other words, roGFP oxidation distinguished within an asynchronous chronologically aging cell culture old mother from young daughter cells.

Authors did a terrific job in isolating old mother cells, by genetic enrichment that select against young daughter cells, and now can establish correlations between replicative, chronologic life span and probe oxidation (Figure 5).

The 24 h data clearly indicates that roGFP oxidation follows RLS. Of the 72-h data, there is a superposition of RLS and CLS: since only mother cells are selected, those that still have a sufficient replicative potential, will therefore inexorably age replicatively, which is what shows Figure 5E, and therefore oxidize (according to the 24 h data). The oxidation shift observed in old cells at 72 h, relative to 24 h might be a consequence of CLS of old, already oxidized cells. Probe oxidation during chronologic aging is anyhow nicely shown in Figure 2.

In conclusion, the redox heterogeneity of a bulk cell culture must be, at least in a large part, contributed to by differences in replicative aging, a conclusion which should be applied to the all study: it is indeed delusive to consider that the 3% reduced cells within 72 h grown old mother cells (Figure 5E) to be representative of the initial reduced and oxidized population used for proteomics and transcriptomics: this might require a bit of reorganization of the paper. Another issue raised by these new data: they are surprisingly totally discrepant with those of Kniess and Mayer (2016), who found a reduction of roGFP2-Grx1 during replicative aging in fermentative medium): authors should definitely address this discrepancy.

2) In the previous evaluation, I asked: "Figure 3C: it is surprising to see that the increase in oxidation during aging gives oxidation values (ratio 405/488) higher than the one measured with diamide (Figure 1D). Please, comment. At best, data should be displayed here using OxD, rather than ratios of fluorescence". Authors nicely introduced OxD values, which is understood as not fitting analyses of heterogeneous cell populations. However, they did not answer the first question: at best, in Figure 2C (former 3C), authors should show the abundance/405/488 ratio graphs for the control DTT and diamide, since they must have been done to calculate the OxDs of Figure 2A.

Another concern I had: "The total probe oxidation of ddh1D at 96 h is very surprising: are these cells dying? Please comment". This tremendous oxidation, equal to the one seen with the diamide control, is extremely surprising. Accordingly, to convince readers, and also avoid any experimental caveats, it seems appropriate to directly measure the levels of GSH and GSSG in whole cells by enzymatic assay, in the same conditions as those of figure 2A (at least time 0 and 72 h) and Figure 7A. It should be acknowledged that this is not the best control, but the only available: whole cells monitoring of GSH and GSSG is biased by the sequestration of GSSG in the vacuole, therefore giving higher levels of GSSG than those actually present in the cytosol (Morgan et al., 2013).

---

## [Author Response]

[Editors’ note: the author responses to the first round of peer review follow.]

Prompted by the comments and suggestions we received, we included additional experimental studies moving beyond the previously described findings, and have significantly expanded our manuscript. Briefly, our revised manuscript addresses all of the points raised by the editors and the reviewers with new data, modified analyses and rephrased text, as is listed below:

The major revisions in the manuscript include:

1) Streamline and focus of the story. Based on all the reviewers’ comments, we have thoroughly reviewed, revised, and expanded figures, figure legends, the supplementary material, and the manuscript. Specifically, in response to Reviewer #2’s suggestions, we have removed the entirety of Figure 2, in order to better focus the current story, while also narrowing the scope of Figure 3 (now Figure 2) in order to increase the clarity of our findings. Additionally, in response to Reviewer #3’s comments, we have extensively adjusted our figures and manuscript to demonstrate the relations between different differentially regulated proteins in the oxidized and reduced subpopulations, moving significant portions to the supplementary material. Moreover, we more carefully rephrased the conclusion, as suggested by the reviewers to focus the story and avoid bold statements.

2) Clarification and distinction between replicative and chronological aging. In response to Reviewer #2’s remarks, we conducted replicative aging experiments (using the Mother Enrichment Program) (Figure 5). Moreover, we linked chronological and replicative aging by comparing the redox status of cells that have undergone multiple replication events with those that have undergone fewer events, at two chronological time points. While there were differences between the two time-points, the highly-divided cells nonetheless displayed a significantly stronger enrichment for oxidation at both time points. From this, we were able to conclude that there is a strong correlation between replication and redox status, though not an exclusive one, while chronological growth also impacts oxidation to a moderate degree. Importantly, we observed that despite undergoing many replications, a small subpopulation of the mother cells remains “eternally reduced”, with a complementary subpopulation of cells that have divided relatively few times that begin life in an oxidized state. This suggests a strong, though not absolute, correlation between replicative aging and redox state. Furthermore, we observe a shift in the redox ratio between the subpopulation of highly divided cells at early growth as compared to late growth, while the degree of cell divisions itself does not change. This points to additional redox changes unrelated to division and replicative events, correlated instead with chronological aging.

This potential heterogeneity was also observed in imaging of dividing cells during 24 hours. Some cells divided several times and maintained a reduced environment, while others underwent oxidation after a lower number of divisions.

We have modified the text and introduced a new figure accordingly (Figure 5). We thank Reviewer #2 for this suggestion as it clarifies a necessary component of the cells under examination.

3) Respiration status of subpopulations. In response to Reviewer #2’s additional comments, we addressed the question regarding the respiratory status of the oxidized subpopulation as compared with the reduced subpopulation. We found that the oxidized subpopulation has a lower respiration rate than the reduced subpopulation (Figure 4), which correlates nicely with growth, as well as with proteomic and transcriptomic profiles. We modified the text and introduced a new figure accordingly (Figure 4).

4) Expansion of analysis of the proteins identified in the proteomic study (Hsp30, Dhh1 and Pnc1). In response to Reviewer #1’s comments, we expanded our analysis of the proteins identified based on their enrichment in the oxidized or reduced subpopulations (Figure 7, Figure S7-S9). We compared the distribution of the 405nm and 488nm ratios for each strain, observing that the distributions largely tracked with the average OxD. Moreover, we used organelle specific sensors, to examine a potential crosstalk between changes in cytosolic oxidation and redox homeostasis in mitochondria and peroxisome. We found that *∆hsp30* displays a unique bi-modal distribution (in the cytosol and mitochondria) at all time-points, with little change in response to chronological aging. On the other hand, the *∆dhh1* strain showed a global change in oxidation (in the cytosol and peroxisome).

Unfortunately, mitochondrial oxidation in the *∆dhh1* was impossible to quantify due to a low fluorescence signal in this strain. We further sought to clarify the differences in their growth by tracking each of these strains under confocal microscopy (Figure 7B), identifying unique growth patterns in accordance with their average characteristics. Interestingly, these two strains showed differences in their morphology, division and growth. While *∆hsp30* has wild-type-like growth and morphology, along with a higher number of oxidized cells than in wild type population, ∆dhh1 led to a more severe phenotype, including slow growth and an enlarged vacuole. Therefore, the effect of these genes is markedly different, with *∆hsp30* affecting the redox-related heterogeneity of cells, shifting cells to more oxidized without influencing their survival. In contrast, deletion of *Dhh1* results in a general collapse of different systems, including redox homeostasis. We thank Reviewer #1 for this helpful suggestion, and we will extend these finding in the future. The text was modified and new figures and videos were included in the manuscript and the supplementary material.

Reviewer #1:

[…] The following points should be taken under considerations to revise the work:- The three proteins identified in the study based on their enrichment in the reduced or oxidized state were functionally examined and concluded to take part in dynamic processes that shape the redox potential of the cell. Hsp30 was in fact implicated earlier. Having established such exciting technology the authors are able to ask more precise questions on which stage these proteins act. Does their absence modify the dynamics of cellular redox changes in the entire population by changing the redox state values of both population groups? Do they affect a bimodal behavior observed in the cell population, i.e. by modifying the number of reduced vs oxidized cells? Does their absence change the ability to enter into the irreversible phase of the oxidized state, or perhaps the daughters of old oxidized mothers are less efficiently recovering into a reduced state?

We thank the reviewer for suggesting that we explore the effect of potentially redox-related proteins, Hsp30, Dhh1 and Pnc1. We extended our analysis of the knockout variants and analyzed the distribution of the cytosolic (Figure 7), mitochondrial and peroxisomal (Figure 7—figure supplement 3, Figure 7—figure supplement 4) oxidation levels within the cell population during chronological aging, while also evaluating the effect of gene deletion on growth (Figure 7—figure supplement 2, Videos 1-4) and cellular morphology (Figure 7B). Moreover, as suggested, we monitored the redox status of the daughter cells of these mutants, with no changes observed relative to wild type cells. Similar to the wild type cells, the daughter cells inherited their mother’s oxidative status. However, a deeper analysis should be done in order to accurately quantify the recovery rate of different knockout strains. For this, we would need to develop a more efficient microscopy-based technology, enabling imaging of a large number of cells, originating from different mutants and detection of the redox status of these cells in either a semi-manual or automatic way.

Results of this analysis are described in the manuscript and summarized below:

“We found *∆dhh1* to be under considerable stress from the onset, with a weaker fluorescent expression and a significantly higher rate of apoptosis as compared to the other strains. […] More specifically, we have identified three proteins that, when knocked out, produce unique oxidation distribution patterns as compared to the wild type.”

- There are plenty of examples, in this study and literature, that the transcriptomics and proteomics frequently do not agree perfectly well. It would interesting to undertake more systematic analyses to see the extent of correlations.

We thank the reviewer for pointing out the partial disagreement between proteomic and transcriptomic data. To address this issue, we conducted a correlation analysis of the mRNA and protein expression levels, plotting and identifying significantly differentially regulated proteins/transcripts across both groups (Figure 8C, Figure 8—figure supplement 2). These plots show that the overwhelming majority of proteins and transcripts are not significantly uncoupled in correlation between transcripts and proteins. However, we do note that the fraction of coupled versus uncoupled transcripts/proteins remains largely the same, suggesting unique regulatory mechanisms for certain transcripts/proteins.

Moreover, we modified the text accordingly:

“In examining the general correlation between the transcriptomic and proteomic analysis at 48h and 72h (Figure 8C, Figure 8—figure supplement 2), we find that while there is a global correlation in the trend (Figure 8C, Figure 8—figure supplement 2), there are several examples of uncoupling in significantly differential protein and mRNA expression (Figure 8C, Figure 8—figure supplement 2, dark red and purple bold circles). While protein expression typically corresponds with mRNA expression, expression of transcripts involved in regulatory roles (Marek and Korona (2013) and aging (Tseng-Rogenski et al. (2003); Chan and Marshall (2014)) have been found to partially decouple.”

- This is somewhat not surprising that the daughter cell inherits the redox status, also the oxidized status, from the mother. An interesting and surprising thing is however that some daughter cells may retain the ability to recover as briefly mentioned at the end of the Results section. This part is somehow slightly underdeveloped in the manuscript. Could the authors develop this part further?

We agree that this part is very interesting and needs further investigation. For this, we would need to develop a methodology that will be able to image (using confocal microscopy) and sort cells based on ability to recover, i.e. reduce oxidation levels in real time. This methodology should be able to deal with relatively large number of cells enabling biochemical or genetic analysis. Until now, we have manually followed dozens of cells and quantified their OxD levels AFTER the imaging experiment has been completed, leaving a majority of cells unsorted, with distinct oxidation status. Analysis of the wild type cells took an enormous amount of manual work, and resulted in a relatively small subset of data.

We completely agree that this is a very interesting finding, and therefore, decided to include it in the paper, even in its undeveloped stage. We hope that these findings will be picked up by other researchers, perhaps with state-of-the-art microfluidic methodologies, for future investigation.

- The authors should avoid bold statements. As already mentioned, the results are largely of correlative nature, and causes and consequences are unclear. For example, I do not see the basis for the following statement: "…we suggest that the redox status, rather than cell age, is a critical determinant of cellular growth and division properties". The text should be carefully evaluated to avoid the statements that do not have a source in literature or in the authors data.

We thank the reviewer for this comment. We have attempted to reduce the number of bold statements to more accurately reflect our results and conclusions.

Reviewer #2:

[…] *There is, however, a concern regarding the interpretation of the data of Figure 4, on which the paper's conclusion is built up. Authors distinguished within a yeast population undergoing chronological aging two subpopulations with a different roGFP2 redox state. Upon inspection of these populations, they found that the reduced one had very few bud scars (< 1), and the oxidized one 3-4.5 more scars: the question is whether the two subpopulations differ by their replicative aging status, with those remaining reduced being newly born daughter cells, and those that oxidize being mother cells, which already underwent a few division, which would be consistent with their growth phenotype, and with the notion that cells oxidize Grx1-roGFP2 during replicative aging, when grown in a respiratory medium (Kniess and Mayer, 2016)? If this the case would collapse the authors' claim that the differences of the two subpopulations are redox, intrinsically, and not age-dependent.*

There are a few other points that must be addressed, as described below.1) Authors must evaluate the possible difference in replicative aging status between the two cells subpopulations: At best, they could sort old and young cells, based on the number of budscars, as done by Kniess and Mayer, and evaluate the redox state of their cytosol during chronological aging, or simultaneously sort cells according to their redox state and budscar status.

We thank the reviewer for this comment and suggestion. We agree that correlation between oxidation and number of division raises a possibility that a reduced environment is an intrinsic and restricted property of newly born cells, that becomes oxidized during replicative aging. We followed the suggestion of the reviewer and established a replicative aging assay using the mother enrichment program. We were unable to use the method described in the Kniess and Mayer paper, instead utilizing a methodology which is based on enrichment of mother cells using selection against daughter cells. We applied sorting to isolate “young” (low number of bud scars) and “old” mother cells (large number of bud scars) at two different periods of stationary phase: early phase (24 hours of growth) and a prolonged stationary state (72 hours). The rationale behind this assay was that if reduction is indeed a property of newly born cells, we will have a similar, increased oxidation in young and old mother cells, without any dependence on duration of the stationary phase. As seen in Figure 5, the outcomes of this experiment strengthen the correlation described earlier between the cellular redox state and cell divisions, identifying a link between cells that have undergone multiple divisions and a strong tendency towards oxidation, though not an exclusive one. We observe that despite undergoing many replications, a small subpopulation of the mother cells remains “eternally reduced”, while there is a complementary subpopulation of cells that have divided relatively few times that begins life in an oxidized state. This suggests a strong, though not absolute, correlation between replicative aging and redox state.

Furthermore, we observe a shift in the redox ratio between the subpopulation of highly divided cells at early growth as compared to late growth, while the degree of cell divisions itself does not change. This points to additional redox changes unrelated to division and replicative events, correlated instead with chronological aging.

2) Table 1 indicate the presence of 30% of reduced cell undergoing budding: It would seem however that chronological aging is the survival of a post-replicative stationary culture: please comment.

We thank the reviewer for the useful comment. To the best of our knowledge, and with agreement with other studies, chronological life span is defined as a period of time that nondividing yeast cells can survive. It is typically defined after entrance to the postdiauxic phase, which starts ~24 hours after initial inoculation, depending on medium and prolongs for few days depending on the growth conditions. During this stage, we and others had observed slow but detectable growth, which might result in these budding events shown in Table 1.

Therefore, in order to correlate more precisely between replication events and oxidation, we utilized an alternative model for aging, the replicative aging model, as described previously. It is important to note, that using the mother enrichment method we clearly detect replication at 48 and 72 hours of growth after the initial inoculation, suggesting that our finding in Table 1 are valid, though this may follow the significantly more diluted sample. We agree that this is not pure “post-replicative stationary phase”, therefore, having an alternative aging model in this study has prominent value, and we thank the reviewer for his suggestion.

3) The mitochondrial status (respiration) should also be established within the two subpopulations.

We thank the reviewer for the suggestion. Respiration was analyzed and presented in Figure 4. Here we found that oxygen consumption rate of the oxidized cells was consistently 1.8-2 fold lower than the reduced population. Decreased oxygen consumption is strongly correlated with reduced cell growth of the oxidized subpopulation vs reduced ones.

4) Figure 3C: it is surprising to see that the increase in oxidation during aging gives oxidation values (ratio 405/488) higher than the one measured with diamide (Figure 1D). Please, comment. At best, data should be displayed here using OxD, rather than ratios of fluorescence.

This is an interesting point. The ratios undergo a shift following natural oxidation of the cells, with all metrics suggesting that these changes lead to higher oxidation ratios than under diamide treatment. This would appear to be a unique expression pattern as a result of endogenous oxidation, clearly separate from the reduced ratios. Furthermore, due to the nature of OxD normalization and changes over time in expression, the distinction between the oxidized and reduced subpopulations would become more difficult to ascertain. Due to the lack of normalization in the 405/488 ratio, we believe this to be the more accurate metric. A clarification regarding this distinction has been added to the manuscript.

5) The values of OxD measured in the cytosol are much higher than published ones using the same probe (Kojer et al., 2012): 0.3 in Wt cytosol (Figure 2A) vs. 0.1. These values are also different in the different figures, which makes the comparison with mutants difficult. In Figure 6E and 7D values in Wt are < 0.1 and those of mutant similar to Wt in 2A. The total probe oxidation of ddh1D at 96 h is very surprising: are these cells dying? Please comment.

This is a good point. We have observed across numerous experiments with the roGFP probes (using these and other strains) a moderate variance in OxD values expressly depending on the culture/colony itself. Due to the fact that the plasmid is easily rejected by cells, we replace the colonies very frequently, and different transformations have subsequently led to different outright expression patterns. Furthermore, because OxD requires new diamide and DTT treatment for every experiment, the normalization is not always identical due to slight variations in treatment duration (between 5-20 minutes). For these reasons, we compared the deletion strains to a new wt colony transformed with them – all colonies from that batch displayed relatively lower OxD values due to some degree to normalization, though their redox ratios remained similar (shown in Figure 7C-F). Regarding the Dhh1 deletion strain, the high oxidation does indeed emerge alongside a high rate of death within the culture; this clarification has been added to the text.

6) The first figures of the paper do not appear really informative and might distract/confuse the reader: Figure 1 displays published data (Kojer et al., 2012). Figure 2 does not provide any data that fit the current story, and use of Glr1 in 3 is not informative because it is known that Glr has a dominant function in reducing glutathione.

We apologize for not being clear regarding the rationale in providing Figures 1-3.

Figure 1 describes our strains expressing the Grx1-roGFP2 sensor, made specifically for this study. The rationale behind presenting this figure was to verify that we obtain similar fluorescence intensity spectrum as in the Kojer et al., 2012 paper as well as to explain the reasoning behind using the 405 and 488nm wavelengths. Therefore, we present here our own data, not published previously. The text was modified to clarify this point.

Regarding Figures 2 and 3, we agree that in the previous version these figures in part did not provide valuable data to fit the story. In the current version, we have removed the original Figure 2, as well as use of Glr1 in the original Figure 3 (now Figure 2). However, we have also decided to use the peroxisomal sensor in order to explore the potential crosstalk between oxidation in the cytosol, mitochondria and peroxisome upon deletion of proteins such as Hsp30 and Dhh1 (Figure 7—figure supplement 4).

Reviewer #3:

[…] However, some of their data analyses do not seem very productive, in particular some of the STRING-based interaction networks and the functional enrichment. A removal of parts of these analyses would probably not affect their overall conclusions. Should they decide to keep these analyses, they need to justify them and describe these analyses in more detail: What were the criteria chosen by the authors to identify an interaction in the interaction networks? What does it e.g. mean if "Hsp30 is connected to Dhh1"? Is it justified to have proteins that were up-regulated in both reduced and oxidized sub-populations in the very same interaction network? What does it mean, if proteins that "interact" are upregulated under diametrically opposed conditions? How were functions defined for the functional enrichment analyses?

We thank the reviewer for this insightful suggestion. We have accordingly removed several of the STRING-based analyses, while transferring the remaining figures to the supplementary material. The criteria for the STRING analysis in each case has been added to the supplementary figure legend and the text has been updated accordingly.

The authors should also submit their proteomics and transcriptomics data to appropriate repositories (e.g. the EMBL PRIDE archive, ArrayExpress, or GEO).

The proteomic and transcriptomic data have been uploaded to the PRIDE archive and GEO database respectively.

[Editors' note: the author responses to the re-review follow.]

[…] We would like to draw your attention to the issue of the oxidation data (the point 2 of reviewer 2). These data should be provided in a full version, and not only as ratios.

Reviewer #2:

1) The heterogeneity in probe oxidation the authors saw during chronologic aging might have been the result of differences in replicative aging, and not just of cellular heterogeneity; in other words, roGFP oxidation distinguished within an asynchronous chronologically aging cell culture old mother from young daughter cells.Authors did a terrific job in isolating old mother cells, by genetic enrichment that select against young daughter cells, and now can establish correlations between replicative, chronologic life span and probe oxidation (Figure 5).The 24 h data clearly indicates that roGFP oxidation follows RLS. Of the 72-h data, there is a superposition of RLS and CLS: since only mother cells are selected, those that still have a sufficient replicative potential, will therefore inexorably age replicatively, which is what shows Figure 5E, and therefore oxidize (according to the 24 h data). The oxidation shift observed in old cells at 72 h, relative to 24 h might be a consequence of CLS of old, already oxidized cells. Probe oxidation during chronologic aging is anyhow nicely shown in Figure 2.In conclusion, the redox heterogeneity of a bulk cell culture must be, at least in a large part, contributed to by differences in replicative aging, a conclusion which should be applied to the all study: it is indeed delusive to consider that the 3% reduced cells within 72 h grown old mother cells (Figure 5E) to be representative of the initial reduced and oxidized population used for proteomics and transcriptomics: this might require a bit of reorganization of the paper.

We agree with the comment that the small subpopulation of reduced cells in mother enriched populations do not necessarily represent the reduced population sorted during chronological aging, in which the reduced cells are a majority. As previously suggested by the reviewer, there is constant division during the stationary phase which contributes to the reduced subpopulation pool. Therefore, it would be interesting, in the future, to use the replicative model for proteomic analysis to understand the origin of the reduced cells in aged mother cells, as well as the oxidized cells in the young mother population. In the meantime, we have modified the manuscript to address this point: “However, it is worth noting that these exceptionally rare cells (in both cases) are representative of the replicative aging model, rather than a chronologically aged culture, in which there appears to be greater turnover in terms of daughter cells and constant replication (including during the stationary phase).”

Another issue raised by these new data: they are surprisingly totally discrepant with those of Kniess and Mayer (2016), who found a reduction of roGFP2-Grx1 during replicative aging in fermentative medium): authors should definitely address this discrepancy.

It is indeed a very interesting paper in which the authors compared the glutathione potential during respiration and fermentation growth, showing that during respiration the aged cells become more oxidized while during fermentation the opposite occurs. Nicely, we also see the same phenomenon in our cells, as grown under aerobic conditions. To emphasize this distinction, we have included the following:

“An interesting example of this may be seen in Knieß and Mayer (Kniess and Mayer, 2016), in which they find a distinction between fermenting and respiratory conditions on glutathione reduction. […] This distinction is interesting in comparison with our results regarding replicative aging under aerobic conditions, opening the door to future exploration.”

2) In the previous evaluation, I asked: "Figure 3C: it is surprising to see that the increase in oxidation during aging gives oxidation values (ratio 405/488) higher than the one measured with diamide (Figure 1D). Please, comment. At best, data should be displayed here using OxD, rather than ratios of fluorescence". Authors nicely introduced OxD values, which is understood as not fitting analyses of heterogeneous cell populations. However, they did not answer the first question: at best, in Figure 2C (former 3C), authors should show the abundance/405/488 ratio graphs for the control DTT and diamide, since they must have been done to calculate the OxDs of Figure 2A.

We thank the reviewer for pointing out the necessary distinction between the 405/488 nm ratios and OxD. Accordingly, we have included an additional figure presenting the distribution of the OxD values during chronological aging (“Figure 2—figure supplement 3) related to the data shown in Figure 2, alongside the existing Figure 1D, which shows the distribution pattern of the DTT and diamide treated cells.

We examined the data extensively in the previous and current versions, to define the best and most mathematically accurate representation of the data. While the OxD is the often-used metric, the necessary averaging of the fluorescence intensity of each cell within the population (for normalization limits) leads to an inaccurate scale and to expanded distributions. While the general trend is preserved, the actual OxD values are not representative (e.g., 150% oxidation).

Moreover, the sorting is done based on the ratio defined gates and not OxD. Therefore, we believe that it is more accurate and intuitive to use the ratio approach than OxD while analyzing cellular variability. In other cases, such as evaluation of the average redox status of a specific population or individual cells under the microscope, the OxD approach is more reflective and clear.

We modified the text accordingly to clarify this point:

“It is important to note that while the average OxD serves as a useful metric by which to analyze yeast populations by flow cytometry, its value diminishes somewhat on an individual, cell-to-cell scale. […] Despite the limitations of using OxD normalization for individual cells of this kind, the distribution shape remains constant.”

Another concern I had: "The total probe oxidation of ddh1D at 96 h is very surprising: are these cells dying? Please comment". This tremendous oxidation, equal to the one seen with the diamide control, is extremely surprising. Accordingly, to convince readers, and also avoid any experimental caveats, it seems appropriate to directly measure the levels of GSH and GSSG in whole cells by enzymatic assay, in the same conditions as those of figure 2A (at least time 0 and 72 h) and Figure 7A. It should be acknowledged that this is not the best control, but the only available: whole cells monitoring of GSH and GSSG is biased by the sequestration of GSSG in the vacuole, therefore giving higher levels of GSSG than those actually present in the cytosol (Morgan et al., 2013).

We agree with the comment and addressed it within the text:

“Further investigation regarding their GSSG/GSH levels, particularly in *∆dhh1* with its enlarged vacuoles (40), would provide greater insight into the mechanism of oxidation/reduction within each strain.”